# DRAUC: An Instance-wise Distributionally Robust AUC Optimization Framework

**Siran Dai**[1,2]        **Qianqian Xu**[3*]        **Zhiyong Yang**[4]
**Xiaochun Cao**[5]        **Qingming Huang**[4,3,6*]

[1] SKLOIS, Institute of Information Engineering, CAS
[2] School of Cyber Security, University of Chinese Academy of Sciences
[3] Key Lab. of Intelligent Information Processing, Institute of Computing Tech., CAS
[4] School of Computer Science and Tech., University of Chinese Academy of Sciences
[5] School of Cyber Science and Tech., Shenzhen Campus of Sun Yat-sen University
[6] BDKM, University of Chinese Academy of Sciences
daisiran@iie.ac.cn        xuqianqian@ict.ac.cn
yangzhiyong21@ucas.ac.cn        caoxiaochun@mail.sysu.edu.cn
qmhuang@ucas.ac.cn

## Abstract

The Area Under the ROC Curve (AUC) is a widely employed metric in long-tailed classification scenarios. Nevertheless, most existing methods primarily assume that training and testing examples are drawn i.i.d. from the same distribution, which is often unachievable in practice. Distributionally Robust Optimization (DRO) enhances model performance by optimizing it for the local worst-case scenario, but directly integrating AUC optimization with DRO results in an intractable optimization problem. To tackle this challenge, methodically we propose an instance-wise surrogate loss of Distributionally Robust AUC (DRAUC) and build our optimization framework on top of it. Moreover, we highlight that conventional DRAUC may induce label bias, hence introducing distribution-aware DRAUC as a more suitable metric for robust AUC learning. Theoretically, we affirm that the generalization gap between the training loss and testing error diminishes if the training set is sufficiently large. Empirically, experiments on corrupted benchmark datasets demonstrate the effectiveness of our proposed method. Code is available at: https://github.com/EldercatSAM/DRAUC.

## 1 Introduction

The Area Under the ROC Curve (AUC) is an essential metric in machine learning. Owing to its interpretation equivalent to the probability of correctly ranking a random pair of positive and negative examples [11], AUC serves as a more suitable metric than accuracy for imbalanced classification problems. Research on AUC applications has expanded rapidly across various scenarios, including medical image classification [40, 51], abnormal behavior detection [5] and more.

However, current research on AUC optimization assumes that the training and testing sets share the same distribution [46], a challenging condition to satisfy when the testing environment presents a high degree of uncertainty. This situation is common in real-world applications.

Distributionally Robust Optimization (DRO) as a technique designed to handle distributional uncertainty, has emerged as a popular solution [38] in various applications, including machine learning [19], energy systems [1] and transportation [25]. This technique aims to develop a model that performs well, even under the most adversarial distribution within a specified distance from the original training distribution. However, existing DRO methods primarily focus on accuracy as a

---

*Corresponding authors.

metric, making it difficult to directly apply current DRO approaches to AUC optimization due to its pairwise formulation. Consequently, it prompts the following question:

***Can we optimize the Distributionally Robust AUC (DRAUC) using an end-to-end framework?***

This task presents three progressive challenges: **1)**: The pairwise formulation of AUC necessitates simultaneous access to both positive and negative examples, which is computationally intensive and infeasible in online settings. **2)**: The naive integration of AUC optimization and DRO leads to an intractable solution. **3)**: Based on a specific observation, we find that the ordinary setting of DRAUC might lead to severe label bias in the adversarial dataset.

In this paper, we address the aforementioned challenges through the following techniques: For **1)**, we employ the minimax reformulation of AUC and present an early trail to explore DRO under the context of AUC optimization. For **2)**, we propose a tractable surrogate loss that is proved to be an upper bound of the original formulation, building our distribution-free DRAUC optimization framework atop it. For **3)**, we further devise distribution-aware DRAUC, to perform class-wise distributional perturbation. This decoupled formulation mitigates the label noise issue. This metric can be perceived as a class-wise variant of the distribution-free DRAUC.

It is worth noting that [56] also discusses the combination of DRO techniques with AUC optimization. However, the scope of their discussion greatly differs from this paper. Their approach focuses on using DRO to construct estimators for partial AUC and two-way partial AUC optimization with convergence guarantees, whereas this paper primarily aims to enhance the robustness of AUC optimization.

The main contributions of this paper include the following:

- **Methodologically**: We propose an approximate reformulation of DRAUC, constructing an instance-wise, distribution-free optimization framework based on it. Subsequently, we introduce the distribution-aware DRAUC, which serves as a more appropriate metric for long-tailed problems.

- **Theoretically**: We conduct a theoretical analysis of our framework and provide a generalization bound derived from the Rademacher complexity applied to our minimax formulation.

- **Empirically**: We assess the effectiveness of our proposed framework on multiple corrupted long-tailed benchmark datasets. The results demonstrate the superiority of our method.

## 2 Related Works

### 2.1 AUC Optimization

AUC is a widely-used performance metric. AUC optimization has garnered significant interest in recent years, and numerous research efforts have been devoted to the field. The researches include different formulations of objective functions, such as pairwise AUC optimization [8], instance-wise AUC optimization [49, 26, 50], AUC in the interested range (partial AUC [48], two-way partial AUC [47]), and area under different metrics (AUPRC [35, 44, 45], AUTKC [43], OpenAUC [42]. For more information, readers may refer to a review on AUC [46].

Some prior work investigates the robustness of AUC. For instance, [52] improves the robustness on noisy data and [15] studies the robustness under adversarial scenarios. In this paper, we further explore robustness under the local worst distribution.

### 2.2 Distributionally Robust Optimization

DRO aims to enhance the robustness and generalization of models by guaranteeing optimal performance even under the worst-case local distribution. To achieve this objective, an ambiguity set is defined as the worst-case scenario closest to the training set. A model is trained by minimizing the empirical risk on the ambiguity set. To quantify the distance between distributions, prior research primarily considers $\phi - divergence$ [2, 16, 4, 31] and the Wasserstein distance [39, 28, 19, 3, 7] as distance metrics. For more details, readers may refer to recent reviews on DRO [28, 23].

DRO has applications in various fields, including adversarial training [39], long-tailed learning [37], label shift [55], etc. However, directly optimizing the AUC on the ambiguity set remains an open problem.

## 3    Preliminaries

In this subsection, we provide a brief review of the AUC optimization techniques and DRO techniques employed in this paper. First, we introduce some essential notations used throughout the paper.

We use $z \in \mathcal{Z}$ to denote the example-label pair, and $f_{\boldsymbol{\theta}} : \mathcal{Z} \to [0, 1]$ to represent a model with parameters $\boldsymbol{\theta} \in \Theta$. This is typical when connecting a Sigmoid function after the model output. For datasets, $\widehat{P}$ denotes the nominal training distribution with $n$ examples, while $P$ represents the testing distribution. We use $\widehat{P}_+ = \{x_1^+, ..., x_{n^+}^+\}$ and $\widehat{P}_- = \{x_1^-, ..., x_{n^-}^-\}$ to denote positive/negative training set, respectively. To describe the degree of imbalance of the dataset, we define $\widehat{p} = \frac{n^+}{n^+ + n^-}$ as the imbalance ratio of training set, and $p = \Pr(y = 1)$ as the imbalance ratio of testing distribution. The notation $\mathbb{E}_P$ signifies the expectation on distribution $P$. We use $c(z, z') = ||z - z'||_2^2$ to denote the cost of perturbing example $z$ to $z'$.

### 3.1    AUC Optimization

Statistically, AUC is equivalent to the Wilcoxon–Mann–Whitney test [11], representing the probability of a model predicting a higher score for positive examples than negative ones

$$AUC(f_{\boldsymbol{\theta}}) = \underset{P_+, P_-}{\mathbb{E}} \left[ \ell_{0,1}(f_{\boldsymbol{\theta}}(\boldsymbol{x}^+) - f_{\boldsymbol{\theta}}(\boldsymbol{x}^-)) \right] \tag{1}$$

where $\ell_{0,1}(\cdot)$ denotes the 0-1 loss, i.e., $\ell_{0,1}(x) = 1$ if $x < 0$ and otherwise $\ell_{0,1}(x) = 0$. Based on this formulation, maximizing AUC is equivalent to the following minimization problem

$$\min_{\boldsymbol{\theta}} \underset{P_+, P_-}{\mathbb{E}} \left[ \ell(f_{\boldsymbol{\theta}}(\boldsymbol{x}^+) - f_{\boldsymbol{\theta}}(\boldsymbol{x}^-)) \right] \tag{2}$$

where $\ell$ is a differentiable, consistent surrogate loss of $\ell_{0,1}$. However, the pairwise formulation of the above loss function is not applicable in an online setting. Fortunately, [49] demonstrates that using the square loss as a surrogate loss, the optimization problem (2) can be reformulated as presented in the following theorem.

**Theorem 1** ([26]). *When using square loss as the surrogate loss, the AUC maximization is equivalent to*

$$\min_{\boldsymbol{\theta}} \underset{P_+, P_-}{\mathbb{E}} \left[ \ell \left( f_{\boldsymbol{\theta}}(\boldsymbol{x}^+) - f_{\boldsymbol{\theta}}(\boldsymbol{x}^-) \right) \right] = \min_{\boldsymbol{\theta}, a, b} \max_{\alpha} \underset{P}{\mathbb{E}} \left[ g(a, b, \alpha, \boldsymbol{\theta}, \boldsymbol{z}) \right] \tag{3}$$

*where*

$$g(a, b, \alpha, \boldsymbol{\theta}, \boldsymbol{z}) = (1 - p) \cdot (f_{\boldsymbol{\theta}}(\boldsymbol{x}) - a)^2 \cdot \mathbb{I}_{[y=1]} + p \cdot (f_{\boldsymbol{\theta}}(\boldsymbol{x}) - b)^2 \cdot \mathbb{I}_{[y=0]}$$
$$+ 2 \cdot (1 + \alpha) \cdot (p \cdot f_{\boldsymbol{\theta}}(\boldsymbol{x}) \cdot \mathbb{I}_{[y=0]} - (1 - p) \cdot f_{\boldsymbol{\theta}}(\boldsymbol{x}) \cdot \mathbb{I}_{[y=1]} - p(1 - p) \cdot \alpha^2). \tag{4}$$

*Moreover, with the parameter $\boldsymbol{\theta}$ fixed, the optimal solution of $a, b, \alpha$, denoted as $a^\star, b^\star, \alpha^\star$, can be expressed as:*

$$a^\star = \underset{P_+}{\mathbb{E}} \left[ f_{\boldsymbol{\theta}}(\boldsymbol{x}^+) \right], b^\star = \underset{P_-}{\mathbb{E}} \left[ f_{\boldsymbol{\theta}}(\boldsymbol{x}^-) \right], \alpha^\star = b^\star - a^\star. \tag{5}$$

**Similar results hold if the true distribution $P_+, P_-$ in the expressions are replaced with $\widehat{P}_+, \widehat{P}_-$.**

**Remark 1** (**The constraints on $a, b, \alpha$**). *Given that the output of the model $f_{\boldsymbol{\theta}}$ is restricted to $[0, 1]$, $a, b, \alpha$ can be confined to the following domains:*

$$\Omega_{a,b} = \{a, b \in \mathbb{R} | 0 \leq a, b, \leq 1\},$$
$$\Omega_{\alpha} = \{\alpha \in \mathbb{R} | -1 \leq \alpha \leq 1\}. \tag{6}$$

*So that the minimax problem can be reformulated as:*

$$\min_{\boldsymbol{\theta}, (a,b) \in \Omega_{a,b}} \max_{\alpha \in \Omega_{\alpha}} \underset{P}{\mathbb{E}} \left[ g(a, b, \alpha, \boldsymbol{\theta}, \boldsymbol{z}) \right]. \tag{7}$$

## 3.2 Distributionally Robust Optimization

Distributionally Robust Optimization (DRO) aims to minimize the learning risk under the local worst-case distribution. Practically, since we can only observe empirical data points, our discussion is primarily focused on empirical distributions. Their extension to population-level is straightforward

$$\min_{\boldsymbol{\theta}} \sup_{\widehat{Q}:d(\widehat{Q},\widehat{P})\leq \epsilon} \mathbb{E}_{\widehat{Q}}[\ell(f_{\boldsymbol{\theta}}, z)] \tag{8}$$

where $\widehat{P}$ is the original empirical distribution, $\widehat{Q}$ is the perturbed distribution and $d$ is the metric of distributional distance. The constraint $d(\widehat{Q}, \widehat{P}) \leq \epsilon$ naturally expresses that the perturbation induced $\widehat{Q}$ should be small enough to be imperceptible.

As demonstrated in [7], when employing the Wasserstein distance $\mathcal{W}_c$ as the metric, a Lagrangian relaxation can be utilized to reformulate DRO into the subsequent minimax problem.

**Theorem 2** ([7]). *With* $\phi_\lambda(z,\boldsymbol{\theta}) = \sup_{z'\in\mathcal{Z}}\{\ell(f_{\boldsymbol{\theta}}, z') - \lambda c(z, z')\}$, *for all distribution* $\widehat{P}$ *and* $\epsilon > 0$, *we have*

$$\sup_{\widehat{Q}:\mathcal{W}_c(\widehat{Q},\widehat{P})\leq \epsilon} \mathbb{E}_{\widehat{Q}}[\ell(f(z))] = \inf_{\lambda\geq 0}\{\lambda\epsilon + \mathbb{E}_{\widehat{P}}[\phi_\lambda(z,\boldsymbol{\theta})]\}. \tag{9}$$

With the theorem above, one can directly get rid of the annoying Wasserstein constraint in the optimization algorithms. We will use this technique to derive an AUC-oriented DRO framework in this paper.

# 4 Method

## 4.1 Warm Up: A Naive Formulation for DRAUC

As a technical warm up, we first start with a straightforward approach to optimize AUC metric directly under the worst-case distribution. By simply incorporating the concept of the Wasserstein ambiguity set, we obtain the following definition of DRAUC in a pairwise style.

**Definition 1** (**Pairwise Formulation of DRAUC**). *Let* $\ell$ *be a consistent loss of* $\ell_{0,1}$, *for any nominal distribution* $\widehat{P}$ *and* $\epsilon > 0$, *we have*

$$DRAUC_\epsilon(f_{\boldsymbol{\theta}}, \widehat{P}) = 1 - \max_{\widehat{Q}:\mathcal{W}_c(\widehat{Q},\widehat{P})\leq \epsilon} \mathbb{E}_{\widehat{Q}}\left[\ell\left(f_{\boldsymbol{\theta}}(\boldsymbol{x}^+) - f_{\boldsymbol{\theta}}(\boldsymbol{x}^-)\right)\right]. \tag{10}$$

However, generating local-worst Wasserstein distribution $\widehat{Q}$ is loss-dependent, implying that we need to know all the training details to deliver a malicious attack. In our endeavor to secure a performance guarantee for our model, we cannot limit the scope of information accessible to an attacker. This pairwise formulation elevates the computational complexity from $O(n)$ to $O(n^+n^-)$, significantly increasing the computational burden. By a simple reuse of the trick in (7), one can immediately reach the following reformulation of the minimization of (10).

**Proposition 1** (**A Naive Reformulation**). *When using square loss as the surrogate loss, The DRAUC minimization problem:* $\min_{\boldsymbol{\theta}} DRAUC_\epsilon(f_{\boldsymbol{\theta}}, \widehat{P})$, *is equivalent to*

$$\textbf{(Ori)} \quad \min_{\boldsymbol{\theta}} \max_{\widehat{Q}:\mathcal{W}_c(\widehat{Q},\widehat{P})\leq \epsilon} \min_{(a,b)\in\Omega_{a,b}} \max_{\alpha\in\Omega_\alpha} \mathbb{E}_{\widehat{Q}}\left[g(a, b, \alpha, \boldsymbol{\theta}, z_i)\right]. \tag{11}$$

Unfortunately, the optimization operators adhere to a min-max-min-max fashion. There is no known optimization algorithm can deal with this kind of problems so far. Hence, in the rest of this section, we will present two tractable formulations as proper approximations of the problem.

## 4.2 DRAUC-Df: Distribution-free DRAUC

Let us take a closer look at the minimax problem ($\boldsymbol{Ori}$). It is straightforward to verify that, fix all the other variables, $g$ is convex with respect to $a, b$ and concave with respect to $\alpha$ within $\Omega_{a,b}, \Omega_\alpha$. We

---

**Algorithm 1** Algorithm for optimizing DRAUC-Df:

---

1: **Input:** the training data $\mathcal{Z}$, step number $K$, step size for inner $K$-step gradient ascent $\eta_z$, learning rates $\eta_\lambda, \eta_w, \eta_\alpha$ and maximal corrupt distance $\epsilon$.
2: **Initialize:** initialize $a^0, b^0, \alpha^0 = 0, \lambda^0 = \lambda_0$.
3: **for** $t = 1$ **to** $T$ **do**
4:     **Sample a batch of example $z$ from $\mathcal{Z}$.**
5:     **Generate Local Worst-Case Examples**:
6:         Initialize $z' = z$.
7:         **for** $k = 1$ **to** $K$ **do**
8:             $z' = \Pi_{\mathcal{Z}}(z' + \eta_z \cdot \nabla_z \phi_{\lambda^t, a, b, \alpha}(\boldsymbol{\theta}, z'))$.
9:         **end for**
10:    **Update Parameters**:
11:        Update $\alpha^{t+1} = \Pi_{\Omega_\alpha}(\alpha^t + \eta_\alpha \cdot \nabla_\alpha g^t(z'))$.
12:        Update $\lambda^{t+1} = \Pi_{\Omega_\lambda}(\lambda^t - \eta_l \cdot \nabla_\lambda[\lambda\epsilon + \phi_{\lambda^t, a, b, \alpha}(\boldsymbol{\theta}, z')])$.
13:        Update $\mathbf{w}^{t+1} = \Pi_{\Omega_\mathbf{w}}(\mathbf{w}^t - \eta_\mathbf{w} \cdot \nabla_\mathbf{w} g^t(z'))$.
14: **end for**

---

are able to interchange the inner $\min_{(a,b) \in \Omega_{a,b}}$ and $\max_{\alpha \in \Omega_\alpha}$ by invoking von Neumann's Minimax theorem [41], which results in

$$\min_{\boldsymbol{\theta}} \max_{\widehat{Q}:\mathcal{W}_c(\widehat{Q},\widehat{P}) \leq \epsilon} \max_{\alpha \in \Omega_\alpha} \min_{(a,b) \in \Omega_{a,b}} \mathbb{E}_{\widehat{Q}}[g(a,b,\alpha,\boldsymbol{\theta},z)]. \tag{12}$$

Moreover, based on the simple property that $\max_x \min_y f(x,y) \leq \min_y \max_x f(x,y)$, we reach an upper bound of the objective function:

$$\underbrace{\max_{\widehat{Q}:\mathcal{W}_c(\widehat{Q},\widehat{P}) \leq \epsilon} \max_{\alpha \in \Omega_\alpha} \min_{(a,b) \in \Omega_{a,b}} \mathbb{E}_{\widehat{Q}}[g(a,b,\alpha,\theta,z)]}_{DRAUC_\epsilon(f_{\boldsymbol{\theta}}, \widehat{P})} \leq \underbrace{\min_{(a,b) \in \Omega_{a,b}} \max_{\alpha \in \Omega_\alpha} \max_{\widehat{Q}:\mathcal{W}_c(\widehat{Q},\widehat{P}) \leq \epsilon} \mathbb{E}_{\widehat{Q}}[g(a,b,\alpha,\theta,z)]}_{\widetilde{DRAUC}_\epsilon(f_{\boldsymbol{\theta}}, \widehat{P})}$$

$$\tag{13}$$

From this perspective, if we minimize $\widetilde{DRAUC}_\epsilon(f_{\boldsymbol{\theta}}, \widehat{P})$ in turn, we can at least minimize an **upper bound** of $DRAUC_\epsilon(f_{\boldsymbol{\theta}}, \widehat{P})$. In light of this, we will employ the following optimization problem as a surrogate for **(Ori)**:

$$(\boldsymbol{Df}) \quad \min_{\mathbf{w}} \max_{\alpha \in \Omega_\alpha} \max_{\widehat{Q}:\mathcal{W}_c(\widehat{Q},\widehat{P}) \leq \epsilon} \mathbb{E}_{\widehat{Q}}[g(\mathbf{w},\alpha,z)] \tag{14}$$

where $\mathbf{w} = \boldsymbol{\theta}, (a,b) \in \Omega_{a,b}$. Now, by applying the strong duality to the inner maximization problem

$$\max_{\widehat{Q}:\mathcal{W}_c(\widehat{Q},\widehat{P}) \leq \epsilon} \mathbb{E}_{\widehat{Q}}[g(\mathbf{w},\alpha,z)]$$

we have

$$(\boldsymbol{Df}) \min_{\mathbf{w}} \max_{\alpha \in \Omega_\alpha} \min_{\lambda \geq 0} \{\lambda\epsilon + \mathbb{E}_{\widehat{P}}[\phi_{\mathbf{w},\lambda,\alpha}(z)]\} \tag{15}$$

where $\phi_{\mathbf{w},\lambda,\alpha}(z) = \max_{z' \in \mathcal{Z}}[g(\mathbf{w},\alpha,z) - \lambda c(z,z')]$. This min-max-min formulation remains difficult to optimize, so we take a step similar to (13) that interchange the inner $\min_{\lambda \geq 0}$ and outer $\max_{\alpha \in \Omega_\alpha}$, resulting in a tractable **upper bound**

$$(\boldsymbol{Df\star}) \min_{\mathbf{w}} \min_{\lambda \geq 0} \max_{\alpha \in \Omega_\alpha} \{\lambda\epsilon + \mathbb{E}_{\widehat{P}}[\phi_{\mathbf{w},\lambda,\alpha}(z)]\}. \tag{16}$$

In this sense, we will use the $(\boldsymbol{Df\star})$ as the final optimization problem for **DRAUC-Df**.

## 4.3 DRAUC-Da: Distribution-aware DRAUC

Though AUC itself is inherently robust toward long-tailed distributions, we also need to examine whether DRAUC shares this resilience. We now present an analysis within a simplified feature space on the real line, where positive and negative examples are collapsed to their corresponding clusters. The choice of the feature space is simple yet reasonable since it is a 1-d special case of the well-accepted neural collapse phenomenon [32, 10, 17, 57, 27].

Specifically, the following proposition states that the distributional attacker in DRAUC can ruin the AUC performance easily by merely attacking the tail-class examples.

**Proposition 2** (**Powerful and Small-Cost Attack on Neural Collapse Feature Space**). *Let the training set comprises $n^+$ positive examples and $n^-$ negative examples in $\mathbb{R}^1$, i.e., $\mathcal{D} = \left\{ x_1^+, ..., x_{n^+}^+, x_{n^++1}^-, ..., x_n^- \right\}$, with the empirical distribution $\widehat{P} = \frac{1}{n} \sum_{i=1}^n \delta_{x_i}$ ($\delta_z$ represents the Dirac point mass at point $z$.). According to the neural collapse assume, we have: $x_i^+ = x^+$, $x_j^- = x^-$. Given a classifier $f(x) = x$, we assume that the maximization of perturb distribution $\widehat{Q}$ is further constrained on the subset:*

$$\mathcal{Q} = \left\{ \widehat{Q} : \widehat{Q} = \frac{1}{n} \sum_{i=1}^n \delta_{x_i'} \right\}$$

*where $x_i \to x_i'$ forms a discrete Monge map. Then, we have:*

$$\inf_{\widehat{Q} \in \mathcal{Q}, AUC(f,\widehat{Q})=0} \mathcal{W}_c(\widehat{P}, \widehat{Q}) \leq \widehat{p} \cdot (1 - \widehat{p}) \cdot (x^+ - x^-)^2$$

*where $\widehat{p} = \frac{n^+}{n}$ is the ratio of the positive examples in the dataset. Moreover, the cost $\widehat{p} \cdot (1 - \widehat{p}) \cdot (x^+ - x^-)^2$ is realized by setting:*

$$x^{+'} = x^{-'} = \widehat{p} \cdot x^+ + (1 - \widehat{p}) \cdot x^-$$

*the barycenter of the two-bodies system $(x^+, x^-)$.*

It is noteworthy that $\widehat{p} \cdot (1 - \widehat{p})$ reflects the degree-of-imbalanceness, which is relatively small for long-tailed datasets. Moreover, the barycenter tends to be pretty close to the head-class examples. Therefore, only the tail-class examples are required to be revised heavily during the attack. In this sense, the attacker can always exploit the tail class examples as a backdoor to ruin the AUC performance with small Wasserstein cost. This is similar to the overly-pessimistic phenomena [6, 16] in DRO. The following example shows how small such cost could be in a numerical sense.

**Example 1.** *Consider a simplified setting in which the training set is comprised of only one positive example and 99 negative examples, i.e., $\widehat{P} = \{x_1^+, x_2^-, ..., x_{100}^-\}$ with $x^+ = 0.99$ and $x^- = 0.01$. The minimum distance required to perturb the AUC metric from 1 to 0 is 0.009702. This result is achieved by perturbing the positive example from 0.99 to 0.0198 and the negative examples from 0.01 to 0.0198, respectively.*

This perturbation strategy indicates a preference towards strong attack on tail-class examples. The resulting distribution $\widehat{Q}$ is always highly biased toward the original distribution, despite the small Wasserstein cost. In the subsequent training process, one has to minimize the expected loss over $\widehat{Q}$, resulting to label noises.

Therefore, it is natural to consider perturbations on the positive and negative distributions separately to avoid such a problem. Accordingly, we propose here a distribution-aware DRAUC formulation:

**Definition 2** (**Distribution-aware DRAUC**). *Let $\ell$ be a consistent loss of $\ell_{0,1}$, for any nominal distribution $\widehat{P}$ and $\epsilon_+, \epsilon_- > 0$, we have*

$$DRAUC_{\epsilon_+,\epsilon_-}^{Da}(f_{\boldsymbol{\theta}}, \widehat{P}) = 1 - \max_{\substack{\widehat{Q}_+ : \mathcal{W}_c(\widehat{Q}_+, \widehat{P}_+) \leq \epsilon_+ \\ \widehat{Q}_- : \mathcal{W}_c(\widehat{Q}_-, \widehat{P}_-) \leq \epsilon_-}} \mathbb{E}_{\widehat{Q}_+, \widehat{Q}_-} \left[ \ell(f_{\boldsymbol{\theta}}(x_i^+) - f_{\boldsymbol{\theta}}(x_j^-)) \right]. \quad (17)$$

For simplicity, let us denote

$$\widehat{\mathcal{Q}} = \{\widehat{Q} | \, \mathcal{W}_c(\widehat{Q}_+, \widehat{P}_+) \leq \epsilon_+, \mathcal{W}_c(\widehat{Q}_-, \widehat{P}_-) \leq \epsilon_-\} \quad (18)$$

Similar to **DRAUC-Df**, we construct our reformulation as follows:

$$\textbf{(Da)} \quad \min_{\mathbf{w}} \max_{\alpha \in \Omega_\alpha} \max_{\widehat{Q} \in \widehat{\mathcal{Q}}} \mathbb{E}_{\widehat{Q}_+, \widehat{Q}_-} [g(a, b, \alpha, \boldsymbol{\theta}, z_i)]. \quad (19)$$

Moreover, we conduct a similar derivation as **DRAUC-Df**, to construct a tractable upper bound:

$$\textbf{(Da⋆)} \min_{\mathbf{w}} \min_{\lambda_+, \lambda_- \geq 0} \max_{\alpha \in \Omega_\alpha} \{\lambda_+ \epsilon_+ + \lambda_- \epsilon_- + \widehat{p} \, \mathbb{E}_{\widehat{P}_+} [\phi_{\mathbf{w},\lambda_+,\alpha}(z)] + (1 - \widehat{p}) \, \mathbb{E}_{\widehat{P}_-} [\phi_{\mathbf{w},\lambda_-,\alpha}(z)]\} \quad (20)$$

where $\phi_{\mathbf{w},\lambda_+,\alpha}(z) = \max_{z' \in \mathcal{Z}}[g(\mathbf{w}, \alpha, z) - \lambda_+ c(z, z')]$ and $\phi_{\mathbf{w},\lambda_-,\alpha}(z) = \max_{z' \in \mathcal{Z}}[g(\mathbf{w}, \alpha, z) - \lambda_- c(z, z')]$. Please see Appendix A for the details.

---

**Algorithm 2** Algorithm for optimizing DRAUC-Da:

---

1: **Input:** the training data $\mathcal{Z}$, step number $K$, step size for inner $K$-step gradient ascent $\eta_z$, learning rates $\eta_\lambda, \eta_w, \eta_\alpha$ and maximal corrupt distance $\epsilon_+, \epsilon_-$.
2: **Initialize:** initialize $a^0, b^0, \alpha^0 = 0, \lambda_+^0 = \lambda_-^0 = \lambda_0$.
3: **for** $t = 1$ **to** $T$ **do**
4:     **Sample a batch of example** $z$ **from** $\mathcal{Z}$.
5:     **Generate Local Worst-Case Examples**:
6:     Initialize $z'_+ = z_+, z'_- = z_-$.
7:     **for** $k = 1$ **to** $K$ **do**
8:         $z'_+ = \Pi_{\mathcal{Z}}(z'_+ + \eta_z \cdot \nabla_z \phi_{\lambda_+^t, a, b, \alpha}(\boldsymbol{\theta}, z'_+))$.
9:         $z'_- = \Pi_{\mathcal{Z}}(z'_- + \eta_z \cdot \nabla_z \phi_{\lambda_-^t, a, b, \alpha}(\boldsymbol{\theta}, z'_-))$.
10:     **end for**
11:     **Update Parameters**:
12:     Update $\alpha^{t+1} = \Pi_{\Omega_\alpha}(\alpha^t + \eta_\alpha \cdot (p\nabla_a g^t(z'_+) + (1-p)\nabla_a g^t(z'_-)))$.
13:     Update $\lambda_+^{t+1} = \Pi_{\Omega_\lambda}(\lambda_+^t - \eta_l \cdot \nabla_{\lambda_+}[\lambda_+\epsilon_+ + \phi_{\lambda_+^t, a, b, \alpha}(\boldsymbol{\theta}, z'_+)])$.
14:     Update $\lambda_-^{t+1} = \Pi_{\Omega_\lambda}(\lambda_-^t - \eta_l \cdot \nabla_{\lambda_-}[\lambda_-\epsilon_- + \phi_{\lambda_-^t, a, b, \alpha}(\boldsymbol{\theta}, z'_-)])$.
15:     Update $\mathbf{w}^{t+1} = \Pi_{\Omega_\mathbf{w}}(\mathbf{w}^t - \eta_\mathbf{w} \cdot (\widehat{p}\nabla_\mathbf{w} g^t(z'_+) + (1-\widehat{p})\nabla_\mathbf{w} g^t(z'_-)))$.
16: **end for**

---

## 4.4 Algorithm

### 4.4.1 DRAUC Optimization

Motivated by the above reformulation, we propose our DRAUC optimization framework, where we solve this optimization problem alternatively.

**Inner maximization problem :** $K$**-step Gradient Ascent**: Following [39], we consider accessing $K$-step gradient ascent with learning rate $\eta_z$ to solve the inner maximization problem, which is widely used in DRO and can be considered as a variance of PGM. For $\alpha$, we use SGA with a step size $\eta_\alpha$.

**Outer minimization problem: Stochastic Gradient Descent**: On each iteration, we apply stochastic gradient descent over $w$ with learning rate $\eta_w$ and over $\lambda$ with learning rate $\eta_\lambda$.

See Algorithms 1,2 for more details.

## 4.5 Generalization Bounds

In this section, we theoretically show that the proposed algorithm demonstrates robust generalization in terms of DRAUC-Da metric, even under local worst-case distributions. That is, we show that a model sufficiently trained under our approximate optimization ($Da\star$) enjoys a reasonable performance guarantee in DRAUC-Da metric. Our analysis based on the standard assumption that the model parameters $\boldsymbol{\theta}$ are chosen from the hypothesis set $\Theta$(such as neural networks of a specific structure). To derive the subsequent theorem, we utilize the results analyzed in Section 4.3 and perform a Rademacher complexity analysis of DRAUC-Da. The proof for DRAUC-Df follows a similar proof and is much simpler, thus we omit the result here. For additional details, please refer to Appendix A.

**Theorem 3 (Informal Version).** *For all* $\boldsymbol{\theta} \in \Theta, \lambda_+, \lambda_- \geq 0, (a, b) \in \Omega_{a,b}, \alpha \in \Omega_\alpha$ *and* $\epsilon_+, \epsilon_- > 0$, *the following inequality holds with a high probability*

$$\underbrace{DRAUC_{\epsilon_+, \epsilon_-}^{Da}(f_{\boldsymbol{\theta}}, P)}_{(a)} \leq \underbrace{\widehat{\mathcal{L}}}_{(b)} + \underbrace{\mathcal{O}(\sqrt{1/\tilde{n}})}_{(c)} \tag{21}$$

*where* $\tilde{n}$ *is some normalized sample size and* $\widehat{\mathcal{L}} = \min_\mathbf{w} \min_{\lambda_+, \lambda_- \geq 0} \max_{\alpha \in \Omega_\alpha} \{\lambda_+\epsilon_+ + \lambda_-\epsilon_- + \widehat{p}\mathbb{E}_{\widehat{P}_+}[\phi_{\mathbf{w}, \lambda_+, \alpha}(z)] + (1 - \widehat{p})\mathbb{E}_{\widehat{P}_-}[\phi_{\mathbf{w}, \lambda_-, \alpha}(z)]\}$.

In Thm.3, $(a)$ represents the robust AUC loss in terms of expectation, $(b)$ denotes the training loss that we use to optimize our model parameters, and $(c)$ is an error term that turns to zero when the

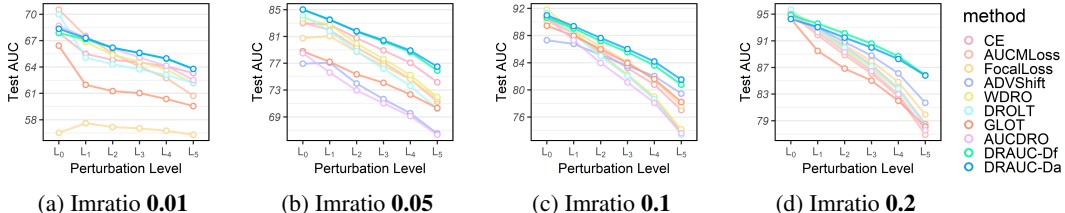

| (a) Imratio **0.01** | (b) Imratio **0.05** | (c) Imratio **0.1** | (d) Imratio **0.2** |

Figure 1: Overall Performance of ResNet32 Across Perturbation Levels on CIFAR10. This graph illustrates the performance of various methods at different corruption levels, with Level 0 indicating no corruption and Level 5 representing the most severe corruption. In each figure, the seven lines depict the test AUC for CE, AUCMLoss, FocalLoss, ADVShift, WDRO, DROLT, GLOT, AUCDRO, DRAUC-Da and DRAUC-Df, respectively. Best viewed in colors.

sample size turns to infinity. In this sense, if we train our model sufficiently within a large enough training set, we can achieve a minimal generalization error.

## 5 Experiments

In this section, we demonstrate the effectiveness of our proposed framework on three benchmark datasets with varying imbalance ratios.

### 5.1 Experiment Settings

We evaluate our framework using the following approach. First, we conduct a binary, long-tailed training set. Then, we proceed to train the model on the long-tailed training set with varying imbalance ratios, tune hyperparameters on the validation set, and evaluate the model exhibiting the highest validation AUC on the corrupted testing set. For instance, we train our model on binary long-tailed MNIST [22], CIFAR10, CIFAR100 [18], and Tiny-ImageNet [21], and evaluate our proposed method on the corrupted version of corresponding datasets [30, 13, 14]. Furthermore, we compare our results with multiple competitors including the baseline (CE), typical methods for long-tailed problems [24, 52, 56] and DRO methods [55, 20, 37, 34]. Please see Appendix B for more details.

### 5.2 Results and Analysis

#### 5.2.1 Overall Performance

The overall performances on CIFAR10 and Tiny-ImageNet are presented in Table 1 and Table 2, respectively. We further compare model performances by altering the level of perturbation, with results displayed in Figure 1. Due to the space limitation, we attach results on MNIST and CIFAR100 in Appendix B. Based on these findings, we make the following observations:

**Effectiveness.** Our proposed method outperforms all competing approaches across Corrupted MNIST, CIFAR10, CIFAR100 and Tiny-ImageNet datasets for all imbalance ratios, thereby substantiating its effectiveness. Additionally, our approach exhibits enhanced performance as the level of perturbation intensifies, indicating its robustness in challenging testing scenarios.

**Ablation results.** Given that our method is modified on AUCMLoss [52], the results presented in Figure 1 can be treated as ablation results. Under the same hyperparameters of AUCMLoss, our method exhibits significant improvement over the baseline, indicating enhanced model robustness.

**Advantage of Distribution-awareness.** As presented in Table 1, DRAUC-Da attains higher scores than DRAUC-Df across almost all corrupted scenarios. This supports our hypothesis that a strong attack on tail-class examples can potentially compromise model robustness.

**Performances on non-corrupted data.** Within non-corrupted datasets, our approach continues to exhibit competitive performance under conditions of extreme data imbalance, specifically when the imbalance ratio equals to 0.01. However, with less imbalanced training data, our method may suffer performance degradation, attributable to the potential trade-off between model robustness and clean performance, which is an unavoidable phenomenon in Adversarial Training [54].

Table 1: Overall Performance on CIFAR10-C and CIFAR10-LT with different imbalance ratios and different models. The highest score on each column is shown with **bold**, and we use darker color to represent higher performance.

| Model | Methods | CIFAR10-C | | | | CIFAR10-LT | | | |
|---|---|---|---|---|---|---|---|---|---|
| | | 0.01 | 0.05 | 0.10 | 0.20 | 0.01 | 0.05 | 0.10 | 0.20 |
| ResNet20 | CE | 62.48 | 75.87 | 83.13 | 86.20 | 65.43 | 84.12 | **92.32** | **95.68** |
| | AUCMLoss | 63.93 | 76.77 | 81.75 | 85.26 | **68.88** | 84.74 | 90.97 | 94.40 |
| | FocalLoss | 56.56 | 74.44 | 81.81 | 84.97 | 57.63 | 81.62 | 91.33 | 94.62 |
| | ADVShift | 61.36 | 75.97 | 83.78 | 87.35 | 64.97 | 82.91 | 87.87 | 95.46 |
| | WDRO | 63.19 | 78.90 | 80.59 | 86.02 | 68.80 | **88.54** | 91.04 | 94.04 |
| | DROLT | 59.92 | 77.51 | 81.09 | 86.46 | 60.99 | 85.76 | 91.35 | 95.17 |
| | GLOT | 63.98 | 77.19 | 83.33 | 87.57 | 65.95 | 88.37 | 90.51 | 94.62 |
| | AUCDRO | 63.35 | 76.19 | 81.82 | 85.96 | 67.14 | 84.00 | 90.92 | 94.88 |
| | DRAUC-Df | 65.58 | **80.18** | 85.71 | 88.83 | 68.12 | 86.47 | 90.57 | 94.17 |
| | DRAUC-Da | **66.06** | 80.13 | **85.91** | **89.51** | 68.71 | 84.43 | 90.30 | 93.76 |
| ResNet32 | CE | 64.43 | 78.79 | 83.12 | 86.89 | 66.05 | 84.40 | 90.44 | **95.61** |
| | AUCMLoss | 64.00 | 76.98 | 81.87 | 85.66 | **68.90** | 84.94 | **91.52** | 95.16 |
| | FocalLoss | 56.96 | 76.53 | 83.82 | 87.42 | 58.04 | 82.99 | 91.02 | 95.16 |
| | ADVShift | 55.74 | 72.42 | 83.47 | 88.32 | 56.73 | 79.36 | 87.88 | 94.95 |
| | WDRO | 64.51 | 78.45 | 83.87 | 88.03 | 68.16 | **86.48** | 90.11 | 95.23 |
| | DROLT | 63.66 | 76.71 | 83.93 | 88.42 | 65.40 | 84.68 | 90.11 | 95.51 |
| | GLOT | 62.59 | 77.21 | 83.67 | 87.30 | 64.53 | 82.62 | 89.59 | 94.62 |
| | AUCDRO | 65.10 | 71.23 | 81.45 | 86.23 | 68.69 | 78.51 | 90.67 | 95.07 |
| | DRAUC-Df | 65.44 | 80.27 | 85.70 | **90.62** | 67.11 | 85.03 | 90.63 | 94.86 |
| | DRAUC-Da | **65.50** | **80.57** | **86.25** | 90.15 | 68.51 | 85.03 | 90.98 | 94.27 |

Table 2: Overall Performance on Tiny-ImageNet-C and Tiny-ImageNet-LT with different imbalance ratios and different models. The highest score on each column is shown with **bold**, and we use darker color to represent higher performance.

| Model | Methods | Tiny-ImageNet-C | | | Tiny-ImageNet-LT | | |
|---|---|---|---|---|---|---|---|
| | | Dogs | Birds | Vehicles | Dogs | Birds | Vehicles |
| ResNet20 | CE | 78.46 | 85.19 | 87.53 | 93.72 | 94.49 | 97.72 |
| | AUCMLoss | 77.35 | 85.98 | 82.37 | 93.35 | 94.11 | 97.34 |
| | FocalLoss | 78.34 | 81.48 | 86.55 | 93.25 | 92.87 | 97.66 |
| | ADVShift | 81.20 | 80.94 | 86.65 | 93.70 | 93.53 | 97.66 |
| | WDRO | 82.20 | 85.23 | 85.92 | 94.46 | 95.50 | **98.19** |
| | DROLT | 80.44 | 86.91 | 86.76 | 93.89 | **96.40** | 97.86 |
| | GLOT | 81.96 | 85.89 | 86.80 | **94.67** | 96.14 | 98.05 |
| | AUCDRO | 75.97 | 83.26 | 79.46 | 92.58 | 93.04 | 96.29 |
| | DRAUC-Df | **84.11** | 87.30 | 88.67 | 93.39 | 95.58 | 97.50 |
| | DRAUC-Da | 83.96 | **87.61** | **89.06** | 93.76 | 95.94 | 97.25 |
| ResNet32 | CE | 82.55 | 84.64 | 86.26 | 94.31 | 94.49 | 97.76 |
| | AUCMLoss | 77.25 | 85.20 | 81.12 | 93.19 | 95.19 | 97.57 |
| | FocalLoss | 77.96 | 79.80 | 85.33 | 93.41 | 92.85 | 97.78 |
| | ADVShift | 84.30 | 84.56 | 86.43 | 92.92 | 94.71 | 97.59 |
| | WDRO | 80.08 | 85.58 | 86.94 | 94.39 | 95.51 | 97.67 |
| | DROLT | 79.25 | 85.75 | 86.79 | 91.68 | **96.06** | 97.82 |
| | GLOT | 81.70 | 83.09 | 88.24 | 94.08 | 95.16 | **97.92** |
| | AUCDRO | 78.21 | 80.55 | 85.26 | 91.56 | 93.15 | 96.33 |
| | DRAUC-Df | **85.79** | **88.00** | 88.32 | **94.43** | 95.29 | 97.37 |
| | DRAUC-Da | 84.56 | 87.60 | **88.46** | 94.03 | 95.96 | 97.65 |

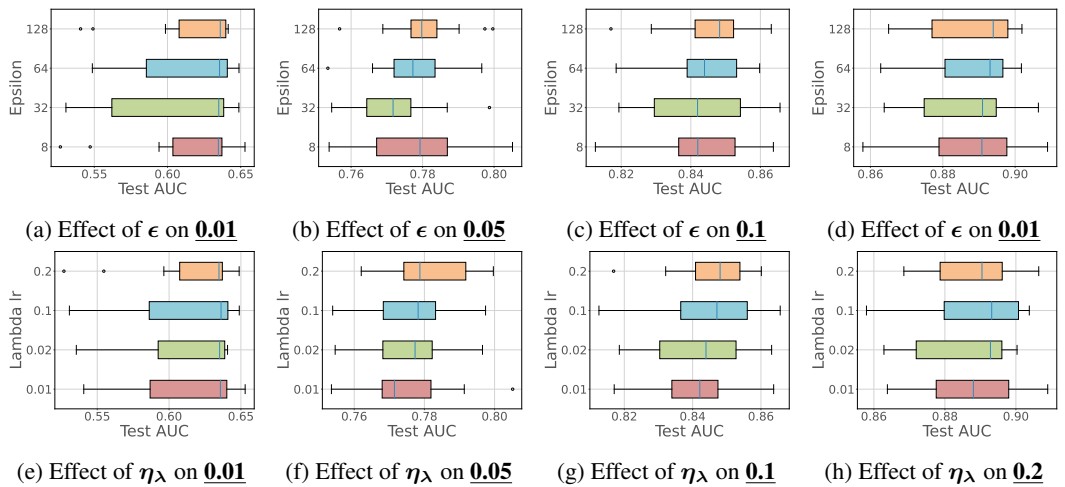

| (a) Effect of $\epsilon$ on **0.01** | (b) Effect of $\epsilon$ on **0.05** | (c) Effect of $\epsilon$ on **0.1** | (d) Effect of $\epsilon$ on **0.01** |
|---|---|---|---|

| (e) Effect of $\eta_\lambda$ on **0.01** | (f) Effect of $\eta_\lambda$ on **0.05** | (g) Effect of $\eta_\lambda$ on **0.1** | (h) Effect of $\eta_\lambda$ on **0.2** |
|---|---|---|---|

Figure 2: Sensitivity analysis of $\epsilon$ and $\eta_\lambda$ on different **imbalance ratios**.

#### 5.2.2 Sensitivity Analysis

**The Effect of $\epsilon$.** In Figure 2-(a)-(d), we present the sensitivity of $\epsilon$. The results demonstrate that when the training set is relatively balanced (i.e., the imbalance ratio $p \geq 0.1$), the average robust performance improves as $\epsilon$ increases. Nonetheless, when the training set is highly imbalanced, the trend is less discernible due to the instability of the training process in these long-tailed settings.

**The Effect of $\eta_\lambda$.** In Figure 2-(e)-(h), we present the sensitivity of $\eta_\lambda$. $\eta_\lambda$ governs the rate of change of $\lambda$ and serves as a similar function to the warm-up epochs in AT. When $\eta_\lambda$ is small, $\lambda$ remains large for an extended period, so the adversarial example is regularized to be less offensive. In cases where the training set is extremely imbalanced, a large $\eta_\lambda$ introduces strong examples to the model while it struggles to learn, increasing the instability of the training process and explaining why the smallest $\eta_\lambda$ performs best with an imbalance ratio of $0.01$. Conversely, when the model does not face difficulty fitting the training data, an appropriately chosen $\eta_\lambda$ around $0.1$ enhances the model's robustness.

## 6 Conclusion and Future Works

This paper presents an instance-wise, end-to-end framework for DRAUC optimization. Due to the pairwise formulation of AUC optimization, a direct combination with DRO is intractable. To address this issue, we propose a tractable surrogate reformulation on top of the instance-wise formulation of AUC risk. Furthermore, through a theoretical investigation on the neural collapse feature space, we find that the distribution-free perturbation is a scheme that might induce heavy label noise into the dataset. In this sense, we propose a distribution-aware framework to handle class-wise perturbation separately. Theoretically, we show that the robust generalization error is small if both the training error and $(1/\sqrt{\tilde{n}})$ is small. Finally, we conduct experiments on three benchmark datasets employing diverse model structures, and the results substantiate the superiority of our approach.

Owing to space constraints, not all potential intersections between AUC optimization and distributionally robustness can be exhaustively explored in this paper. Numerous compelling aspects warrant further investigation. We offer a detailed, instance-wise reformulation of DRAUC, primarily evolving from an AUC optimization standpoint. Future discussions could benefit from initiating dialogue from the angle of DRO. Additionally, integrating various formulations of AUC such as partial AUC and AUPRC with distributional robustness presents a fertile ground for exploration. The existence of a potentially overly-pessimistic phenomenon is yet to be conclusively determined, which paves the way for future inquiries and discoveries.

## Acknowledgements

This work was supported in part by the National Key R&D Program of China under Grant 2018AAA0102000, in part by National Natural Science Foundation of China: 62236008, U21B2038, U2001202, 61931008, 62122075, 61976202, and 62206264, in part by the Fundamental Research Funds for the Central Universities, in part by Youth Innovation Promotion Association CAS, in part by the Strategic Priority Research Program of Chinese Academy of Sciences (Grant No. XDB28000000) and in part by the Innovation Funding of ICT, CAS under Grant No. E000000.

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
