# Appendices

## Contents

# A   Proofs

## A.1   Proof of Proposition 2

*Proof.* We first give a description of the problem. Our objective is to identify the corrupted distribution that minimizes the Wasserstein distance to the original distribution, while simultaneously perturbing the AUC from 1 to 0. Specifically,

$$\min \mathcal{W}_c(\widehat{Q}, \widehat{P}) \tag{22}$$

$$s.t. AUC(f_{\boldsymbol{\theta}}, \widehat{Q}) = 0 \tag{23}$$

From the definition of Wasserstein distance, we have

$$\mathcal{W}_c(\widehat{P}, \widehat{Q}) = \min_{\boldsymbol{\Gamma}} \sum_{i=1}^{n} \sum_{j=1}^{n} \Gamma_{i,j} c_x(z_i, z_j') \tag{24}$$

$$s.t. \quad \Gamma_{i,j} \geq 0, \boldsymbol{\Gamma}^T \mathbf{1} = \frac{1}{n} \mathbf{1}, \boldsymbol{\Gamma} \mathbf{1} = \frac{1}{n} \mathbf{1}, AUC(f_{\boldsymbol{\theta}}, \widehat{Q}) = 0 \tag{25}$$

where $\boldsymbol{\Gamma}$ is the optimal transportation matrix between $\widehat{P}, \widehat{Q}$ and $c_x(z, z') = (x - x')^2 + \infty \cdot \mathbb{I}(y \neq y')$ is a metric of distance between sample $z$ and $z'$.

**Step 1): Separating positive and negative distance.** From the definition of $c_x$, it is easy to check that $c_x(z_i, z_j') = \infty$ if $i \leq n^+, j > n^+$ or $i > n^+, j \leq n^+$. Consequently, the Wasserstein distance goes infinity if $\Gamma_{i,j} > 0$ in the corresponding area, resulting in

$$\boldsymbol{\Gamma} = \begin{bmatrix} \begin{matrix} \Gamma_{1,1} & \cdots & \Gamma_{1,n^+} \\ \vdots & \ddots & \vdots \\ \Gamma_{n^+,1} & \cdots & \Gamma_{n^+,n^+} \end{matrix} & \mathbf{0} \\ \mathbf{0} & \begin{matrix} \Gamma_{n^++1,n^++1} & \cdots & \Gamma_{n^++1,n} \\ \vdots & \ddots & \vdots \\ \Gamma_{n,1} & \cdots & \Gamma_{n,n} \end{matrix} \end{bmatrix} = \begin{bmatrix} \boldsymbol{\Gamma}^+ & 0 \\ 0 & \boldsymbol{\Gamma}^- \end{bmatrix} \tag{26}$$

Now, we can rewrite the Wasserstein distance by separating positive and negative examples

$$\mathcal{W}_c(\widehat{P}, \widehat{Q}) = \min_{\boldsymbol{\Gamma}} \underbrace{\sum_{i=1}^{n_+} \sum_{j=1}^{n_+} \Gamma_{i,j}(x_i^+ - x_j^{+'})^2}_{positive} + \underbrace{\sum_{i=n_++1}^{n} \sum_{j=n_++1}^{n} \Gamma_{i,j}(x_i^- - x_j^{-'})^2}_{negative} \tag{27}$$

$$s.t. \quad \Gamma_{i,j} \geq 0, \boldsymbol{\Gamma}^T \mathbf{1} = \frac{1}{n} \mathbf{1}, \boldsymbol{\Gamma} \mathbf{1} = \frac{1}{n} \mathbf{1}, AUC(f_{\boldsymbol{\theta}}, \widehat{Q}) = 0 \tag{28}$$

**Step 2): Cancelling $\boldsymbol{\Gamma}$.** Plugging in $x_i^+ = x^+, x_j^- = x^-, \forall i, j$, yields the Wasserstein distance of positive class can be considered as

$$\sum_{i=1}^{n_+} \sum_{j=1}^{n_+} \Gamma_{i,j}(x_i^+ - x_j^{+'})^2 \tag{29}$$

$$= \sum_{j=1}^{n_+} \left( \sum_{i=1}^{n_+} \Gamma_{i,j}(x^+ - x_j^{+'})^2 \right) \tag{30}$$

$$= \sum_{j=1}^{n_+} \left( \sum_{i=1}^{n_+} \Gamma_{i,j} \right) (x^+ - x_j^{+'})^2 \tag{31}$$

$$= \sum_{j=1}^{n_+} \frac{1}{n}(x^+ - x_j^{+'})^2 \tag{32}$$

Taking a similar step toward the negative Wasserstein distance, yields that

$$\mathcal{W}_c(\widehat{P}, \widehat{Q}) = \sum_{j=1}^{n_+} \frac{1}{n}(x^+ - x_j^{+\prime})^2 + \sum_{j=n^++1}^{n} \frac{1}{n}(x^- - x_j^{-\prime})^2 \tag{33}$$

Hence, we only need to analysis the problem:

$$\min_{\boldsymbol{x}^{+\prime}, \boldsymbol{x}^{-\prime}} \sum_{j=1}^{n_+} \frac{1}{n}(x^+ - x_j^{+\prime})^2 + \sum_{j=n^++1}^{n} \frac{1}{n}(x^- - x_j^{-\prime})^2 \tag{34}$$

$$s.t. \quad \max \boldsymbol{x}^{+\prime} \leq \min \boldsymbol{x}^{-\prime} \tag{35}$$

where $\boldsymbol{x}^{+\prime} = \{x_1^{+\prime}, ..., x_{n^+}^{+}{}^{\prime}\}, \boldsymbol{x}^{-\prime} = \{x_{n^++1}^{-}{}^{\prime}, ..., x_n^{-\prime}\}$. The constraint comes from the definition of AUC [11].

**Step 3): Solving the optimal perturbations.**

We now show that, the optimal $\boldsymbol{x}^{+\prime}, \boldsymbol{x}^{-\prime}$ consists of same element, and we construct the proof by contradiction. Assume that the optimal perturbation of positive class $\boldsymbol{x}^{+,\star}$ and $\boldsymbol{x}^{-,\star}$. For the positive examples, we assume that the vector $\boldsymbol{x}^{+,\star}$ has at least two different values. Moreover, we check the simple solution $\widehat{x}^{+,\star}$ such that:

$$\widehat{x}^{+,\star} = \arg\min_{x \in \boldsymbol{x}^{+,\star}}(x^+ - x)^2$$

and denote $\widehat{\boldsymbol{x}}^{+\prime} = \{\widehat{x}^{+,\star}, ..., \widehat{x}^{+,\star}\}$. It is easy to check that

$$\sum_{j=1}^{n_+} \frac{1}{n}(x^+ - \widehat{x}^{+,\star})^2 \leq \sum_{j=1}^{n_+} \frac{1}{n}(x^+ - x_j^{+\prime})^2 \tag{36}$$

Furthermore, since

$$\max \widehat{\boldsymbol{x}}^{+\prime} \leq \max \boldsymbol{x}^{+,\star} \leq \min \boldsymbol{x}^{-,\star}, \tag{37}$$

we see that $\widehat{\boldsymbol{x}}^{+\prime}$ is also a feasible solution of the problem. Hence, $\widehat{\boldsymbol{x}}^{+\prime}$ should be the optimal solution instead of $\boldsymbol{x}^{+,\star}$. Following a similar spirit, we can also show that $\boldsymbol{x}^{-,\star}$ is not the optimal solution. In this since, the optimal solution of both $\boldsymbol{x}^{+\prime}$ and $\boldsymbol{x}^{-\prime}$ must be a vector containing the same value.

In this sense, we can further simplfy the targeted optimization problem as:

$$\min_{x^{+\prime}, x^{-\prime}} \widehat{p}(x^+ - x^{+\prime})^2 + (1 - \widehat{p})(x^- - x^{-\prime})^2 \tag{38}$$

$$s.t. \quad x^{+\prime} \leq x^{-\prime} \tag{39}$$

where $\widehat{p} = \frac{n^+}{n}$ is the ratio of the positive examples in the dataset.

**Step 4): Calculating an upper bound of the objective function.** To obtain an upper bound, we can instead check the solution of the following problem:

$$\min_{x'} \widehat{p}(x^+ - x')^2 + (1 - \widehat{p})(x^- - x')^2 \tag{40}$$

It achieves an upper bound since $x^{+\prime} \leq x^{-\prime}$ is automatically satisfied by setting $x^{+\prime} = x^{-\prime} = x'$. By solving this problem, we can see that the optimal solution is:

$$x' = \widehat{p}x^+ + (1 - \widehat{p})x^-, \tag{41}$$

and an upper bound of minimal Wasserstein distance to perturb AUC from 1 to 0 is $\widehat{p}(1 - \widehat{p})(x^+ - x^-)^2$. $\qquad\square$

## A.2 Derivations of Optimization Problem (20)

**Remark 2.** *The original optimization of Distribution-aware DRAUC is*

$$(\mathbf{DRAUC - Da}) \quad \min_{\boldsymbol{\theta}} \max_{\widehat{Q} \in \widehat{\mathcal{Q}}} \min_{(a,b) \in \Omega_{a,b}} \max_{\alpha \in \Omega_\alpha} \mathop{\mathbb{E}}_{\widehat{Q}_+, \widehat{Q}_-} [g(a, b, \alpha, \boldsymbol{\theta}, z_i)]. \tag{42}$$

Similar to what we have done in Section 4.2, for a fixed $\boldsymbol{\theta}, \widehat{Q}_+, \widehat{Q}_-$, we are able to interchange the inner $\min_{(a,b)\in\Omega_{a,b}}$ and $\max_{\alpha\in\Omega_\alpha}$ by invoking von Neumann's Minimax theorem [41], which results in

$$\min_{\boldsymbol{\theta}} \max_{\widehat{Q}\in\widehat{\mathcal{Q}}} \max_{\alpha\in\Omega_\alpha} \min_{(a,b)\in\Omega_{a,b}} \mathbb{E}_{\widehat{Q}_+,\widehat{Q}_-} [g(a,b,\alpha,\boldsymbol{\theta},z)]. \tag{43}$$

Subsequently, based on the property that $\max_x \min_y f(x,y) \le \min_y \max_x f(x,y)$, we reach an upper bound of the objective function:

$$\underbrace{\max_{\widehat{Q}\in\widehat{\mathcal{Q}}} \max_{\alpha\in\Omega_\alpha} \min_{(a,b)\in\Omega_{a,b}} \mathbb{E}_{\widehat{Q}_+,\widehat{Q}_-} [g(a,b,\alpha,\theta,z)]}_{DRAUC^{Da}_{\epsilon_+,\epsilon_-}(f_{\boldsymbol{\theta}},\widehat{P})} \le \underbrace{\min_{(a,b)\in\Omega_{a,b}} \max_{\alpha\in\Omega_\alpha} \max_{\widehat{Q}\in\widehat{\mathcal{Q}}} \mathbb{E}_{\widehat{Q}_+,\widehat{Q}_-} [g(a,b,\alpha,\theta,z)]}_{\widetilde{DRAUC}^{Da}_{\epsilon_+,\epsilon_-}(f_{\boldsymbol{\theta}},\widehat{P})} \tag{44}$$

From this perspective, if we minimize $\widetilde{DRAUC}^{Da}_{\epsilon_+,\epsilon_-}(f_{\boldsymbol{\theta}},\widehat{P})$ in turn, we can at least minimize an **upper bound** of $DRAUC^{Da}_{\epsilon_+,\epsilon_-}(f_{\boldsymbol{\theta}},\widehat{P})$. In light of this, we will employ the following optimization problem as a surrogate for $(\mathbf{DRAUC} - \mathbf{Da})$:

$$(\boldsymbol{Da}) \quad \min_{\mathbf{w}} \max_{\alpha\in\Omega_\alpha} \max_{\widehat{Q}\in\widehat{\mathcal{Q}}} \mathbb{E}_{\widehat{Q}_+,\widehat{Q}_-} [g(\mathbf{w},\alpha,z)] \tag{45}$$

where $\mathbf{w} = \boldsymbol{\theta}, (a,b) \in \Omega_{a,b}$. To further derive a simplified upper bound, one should note that

$$\mathbb{E}_{\widehat{Q}_+,\widehat{Q}_-} [g(\mathbf{w},\alpha,z)] = \widehat{p} \mathbb{E}_{\widehat{Q}_+} [g(\mathbf{w},\alpha,z)] + (1-\widehat{p}) \mathbb{E}_{\widehat{Q}_-} [g(\mathbf{w},\alpha,z)]$$

Hence the inner maximization admits an upper bound:

$$\max_{\widehat{Q}\in\widehat{\mathcal{Q}}} \mathbb{E}_{\widehat{Q}_+,\widehat{Q}_-} [g(\mathbf{w},\alpha,z)] \le \widehat{p} \max_{\widehat{Q}_+\le\epsilon_+} \mathbb{E}_{\widehat{Q}_+} [g(\mathbf{w},\alpha,z)] + (1-\widehat{p}) \max_{\widehat{Q}_-\le\epsilon_-} \mathbb{E}_{\widehat{Q}_-} [g(\mathbf{w},\alpha,z)]$$

By adopting Thm.2, we reach the correspding upper bound:

$$(\boldsymbol{Da}) = \min_{\mathbf{w}} \max_{\alpha\in\Omega_\alpha} \min_{\lambda_+,\lambda_-\ge0} \{\lambda_+\epsilon_+ + \lambda_-\epsilon_- + \widehat{p} \mathbb{E}_{\widehat{P}_+} [\phi_{\mathbf{w},\lambda_+,\alpha}(z)] + (1-\widehat{p}) \mathbb{E}_{\widehat{P}_-} [\phi_{\mathbf{w},\lambda_-,\alpha}(z)]\} \tag{46}$$

where $\phi_{\mathbf{w},\lambda,\alpha}(z) = \max_{z'\in\mathcal{Z}}[g(\mathbf{w},\alpha,z) - \lambda c(z,z')]$. This min-max-min formulation remains difficult to optimize, so we take a step similar to (13) that interchange the inner $\min_{\lambda_+,\lambda_-\ge0}$ and outer $\max_{\alpha\in\Omega_\alpha}$, resulting in a tractable **upper bound**

$$(\boldsymbol{Da\star}) \min_{\mathbf{w}} \min_{\lambda_+,\lambda_-\ge0} \max_{\alpha\in\Omega_\alpha} \{\lambda_+\epsilon_+ + \lambda_-\epsilon_- + \widehat{p} \mathbb{E}_{\widehat{P}_+} [\phi_{\mathbf{w},\lambda_+,\alpha}(z)] + (1-\widehat{p}) \mathbb{E}_{\widehat{P}_-} [\phi_{\mathbf{w},\lambda_-,\alpha}(z)]\} \tag{47}$$

### A.3 Proof of Theorem 3

Since we optimize DRAUC-Da in a class-wise manner, we now give our definition of Rademacher Complexity on positive dn negative distributions, respectively.

**Definition 3** (**Definition of Rademacher Complexity of Robust AUC**). *Given a hypothesis class $\Theta$ and empirical distribution $\widehat{P}$, for all $t \in \Theta, \lambda_+ \ge 0, \lambda_- \ge 0, \alpha \in \Omega_\alpha, (a,b) \in \Omega_{a,b}$, the Positive/Negative Empirical Rademacher Complexity of Robust AUC is defined as*

$$\widehat{\mathfrak{R}}^+_{\widehat{P}_+}(\Theta) = \mathbb{E}_\sigma \left[ \sup_{\mathbf{w},\lambda_+\ge\mathbf{0},\alpha\in\boldsymbol{\Omega}_\alpha} \frac{1}{n_+} \sum_{i=1}^{n_+} \sigma_i \cdot \phi_{\mathbf{w},\lambda_+,\alpha}(z) \right], \tag{48}$$

$$\widehat{\mathfrak{R}}^-_{\widehat{P}_-}(\Theta) = \mathbb{E}_\sigma \left[ \sup_{\mathbf{w},\lambda_-\ge\mathbf{0},\alpha\in\boldsymbol{\Omega}_\alpha} \frac{1}{n_-} \sum_{i=1}^{n_-} \sigma_i \cdot \phi_{\mathbf{w},\lambda_-,\alpha}(z) \right], \tag{49}$$

*where $\sigma$ is the Rademacher random variable, and Positive/Negative Rademacher Complexity of Robust AUC on hypothesis class $\Theta$ is*

$$\mathfrak{R}^+_m(\Theta) = \mathbb{E}_{\widehat{P}_+} [\widehat{\mathfrak{R}}^+_{\widehat{P}_+}(\Theta)], \mathfrak{R}^-_m(\Theta) = \mathbb{E}_{\widehat{P}_-} [\widehat{\mathfrak{R}}^-_{\widehat{P}_-}(\Theta)]. \tag{50}$$

The main result could be restated formally in the following sense.

**Theorem 4** (**Restate of Theorem 3**). *If the samples of the training drawn `i.i.d.`, then for all $\boldsymbol{\theta} \in \Theta, (a,b) \in \Omega_{a,b}, \alpha \in \Omega_\alpha, \lambda_+ \geq 0, \lambda_- \geq 0$, the following holds with probability at least $1 - \delta$ over the randomness of the sample:*

$$DRAUC_{\epsilon_+,\epsilon_-}^{Da}(f_{\boldsymbol{\theta}}, P) \leq \widehat{\mathcal{L}} + 2 \cdot \widehat{p} \cdot \widehat{\mathfrak{R}}_{\widehat{P}_+}^+ (\Theta) + 2 \cdot (1 - \widehat{p}) \cdot \widehat{\mathfrak{R}}_{\widehat{P}_-}^- (\Theta) +$$

$$C_+ \cdot \widehat{p} \cdot \sqrt{\frac{\log(8/\delta)}{2n_+}} + C_- \cdot (1 - \widehat{p}) \cdot \sqrt{\frac{\log(8/\delta)}{2n_-}} +$$

$$2 \cdot C_\infty \cdot \sqrt{\frac{\log(1/\delta)}{2n}}$$

*where $C_+, C_-, C_\infty$ are universal constants, $p = \mathbb{P}[y = 1], \widehat{p} = \widehat{\mathbb{P}}[y = 1]$ and*

$$\widehat{\mathcal{L}} = \min_{\mathbf{w}} \min_{\lambda_+,\lambda_- \geq 0} \max_{\alpha \in \Omega_\alpha} \left[ \lambda_+ \epsilon_+ + \lambda_- \epsilon_- + \widehat{p} \, \underset{\widehat{P}_+}{\mathbb{E}} \left[ \phi_{\mathbf{w},\lambda_+,\alpha}(z) \right] + (1 - \widehat{p}) \, \underset{\widehat{P}_-}{\mathbb{E}} \left[ \phi_{\mathbf{w},\lambda_-,\alpha}(z) \right] \right]$$

*is the saddle point of the training loss.*

**Remark 3.** *The claim of Thm.3 holds since the Rademacher complexity of training data with size $n$ is known to be scaled like $O(\sqrt{1/n})$ for many hypothesis classes such as linear classifiers [29] and neural networks [9].*

We now give a detailed proof of Theorem 3. As the begining, we give some useful lemmas in proving the result.

**Lemma 1.** *The following inequality holds for all $\boldsymbol{\theta} \in \Theta, \epsilon_+, \epsilon_- > 0$*

$$DRAUC_{\epsilon_+,\epsilon_-}^{Da}(f_{\boldsymbol{\theta}}, P) \tag{51}$$

$$\leq \min_{(a,b) \in \Omega_{a,b}} \min_{\lambda_+,\lambda_- \geq 0} \max_{\alpha \in \Omega_\alpha} \left\{ \lambda_+ \epsilon_+ + \lambda_- \epsilon_- + p \, \underset{P_+}{\mathbb{E}} \left[ \phi_{\mathbf{w},\lambda_+,\alpha}(z) \right] + (1 - p) \, \underset{P_-}{\mathbb{E}} \left[ \phi_{\mathbf{w},\lambda_-,\alpha}(z) \right] \right\} \tag{52}$$

*Proof.* The proof is the similar to the proof derivations in the last subsection, except dropping the outer $\min_\theta$ and changing the empirical distribution $\widehat{P}$ to the real distribution $P$. $\qquad\square$

**Lemma 2.** *For any real valued function continuous function: $f : \mathbb{R} \to \mathbb{R}$, $g : \mathbb{R} \to \mathbb{R}$, and for any tight set $\mathcal{X} \subset \mathbb{R}$:*

$$\max_{x \in \mathcal{X}} f(x) - \max_{x' \in \mathcal{X}} g(x') \leq \max_{x \in \mathcal{X}} f(x) - g(x)$$

$$\min_{x \in \mathcal{X}} f(x) - \min_{x' \in \mathcal{X}} g(x') \leq \max_{x \in \mathcal{X}} f(x) - g(x)$$

*Proof.* Since both $f$ and $g$ are continuous, and $\mathcal{X}$ is tight, we now that the maximum and the minimum in the lemma exists. From the basic property of the maxima, we have:

$$\max_{x \in \mathcal{X}} f(x) - \max_{x' \in \mathcal{X}} g(x') = \max_{x \in \mathcal{X}} \min_{x' \in \mathcal{X}} f(x) - g(x') \leq \max_{x \in \mathcal{X}} f(x) - g(x').$$

Similarly, for the minimum, we have:

$$\min_{x \in \mathcal{X}} f(x) - \min_{x' \in \mathcal{X}} g(x') = \min_{x \in \mathcal{X}} \max_{x' \in \mathcal{X}} f(x) - g(x')$$

$$= \max_{x' \in \mathcal{X}} \min_{x \in \mathcal{X}} f(x) - g(x') \leq \max_{x \in \mathcal{X}} f(x) - g(x).$$

$\square$

**Lemma 3.** *Assume that for each $\boldsymbol{x} \in \mathcal{X}$, there exist a sample pair $(\boldsymbol{x}, \boldsymbol{x}') \in \mathcal{X} \times \mathcal{X}$, such that $d(\boldsymbol{x}, \boldsymbol{x}') < \infty$, we have the following result holds for the risk function:*

$$\sup_{\boldsymbol{\theta} \in \Theta} \Bigg[ DRAUC_{\epsilon_+, \epsilon_-}^{Da}(f_{\boldsymbol{\theta}}, P)$$

$$- \min_{(a,b) \in \Omega_{a,b}} \min_{\lambda_+, \lambda_- \geq 0} \max_{\alpha \in \Omega_\alpha} \left\{ \lambda_+ \epsilon_+ + \lambda_- \epsilon_- + \widehat{p} \cdot \underset{\widehat{P}_+}{\mathbb{E}} \left[ \phi_{\mathbf{w}, \lambda_+, \alpha}(z) \right] + (1 - \widehat{p}) \cdot \underset{\widehat{P}_-}{\mathbb{E}} \left[ \phi_{\mathbf{w}, \lambda_-, \alpha}(z) \right] \right\} \Bigg]$$

$$\leq \widehat{p} \cdot \sup_{\boldsymbol{\theta} \in \Theta, (a,b) \in \Omega_{a,b}, \lambda_+ \geq 0} \left[ \underset{P_+}{\mathbb{E}} \left[ \phi_{\mathbf{w}, \lambda_+, \alpha}(z) \right] - \underset{\widehat{P}_+}{\mathbb{E}} \left[ \phi_{\mathbf{w}, \lambda_+, \alpha}(z) \right] \right] +$$

$$(1 - \widehat{p}) \cdot \sup_{\boldsymbol{\theta} \in \Theta, (a,b) \in \Omega_{a,b}, \lambda_- \geq 0} \left[ \underset{P_-}{\mathbb{E}} \left[ \phi_{\mathbf{w}, \lambda_-, \alpha}(z) \right] - \underset{\widehat{P}_-}{\mathbb{E}} \left[ \phi_{\mathbf{w}, \lambda_-, \alpha}(z) \right] \right] + 2 \cdot C_\infty \cdot |p - \widehat{p}|$$

*where:*

$$0 \leq C_\infty = \sup_{\boldsymbol{z} \in \mathcal{Z}, \boldsymbol{w}, \alpha \in \Omega_\alpha, \lambda \geq 0} \phi_{\mathbf{w}, \lambda, \alpha}(z) < \infty.$$

*Proof.* First, we proof the claim that:

$$0 \leq C_\infty < \infty$$

We have:

$$\max_{z' \in \mathcal{Z}, \mathbf{w}, \alpha \in \Omega_\alpha, \lambda \geq 0} [g(\mathbf{w}, \alpha, z) - \lambda c(z, z')] \geq \max_{\mathbf{w}, \alpha \in \Omega_\alpha, \lambda \geq 0} [g(\mathbf{w}, \alpha, z) - \lambda c(z, z)] = \max_{\mathbf{w}, \alpha \in \Omega_\alpha} [g(\mathbf{w}, \alpha, z)].$$

Moreover, since the output of the scoring function resides in $[0, 1]$, we know that $g(\mathbf{w}, \alpha, z)$ is bounded from below uniformly by 0. Hence, $0 \leq C_\infty$.

Similarly,

$$\max_{z' \in \mathcal{Z}, \mathbf{w}, \alpha \in \Omega_\alpha, \lambda \geq 0} [g(\mathbf{w}, \alpha \in \Omega_\alpha, z) - \lambda c(z, z')] \leq \max_{\mathbf{w}, \alpha} [g(\mathbf{w}, \alpha, z)]$$

since $c(z, z) \geq 0$. Moreover, since the output of the scoring function resides in $[0, 1]$, we know that $g(\mathbf{w}, \alpha, z)$ is bounded from above uniformly by some finite constant $B < \infty$. Hence, $C_\infty < \infty$.

From Lem.1, we have:

$$\sup_{\boldsymbol{\theta} \in \Theta} \Bigg[ DRAUC_{\epsilon_+, \epsilon_-}^{Da}(f_{\boldsymbol{\theta}}, P)$$

$$- \min_{(a,b) \in \Omega_{a,b}} \min_{\lambda_+, \lambda_- \geq 0} \max_{\alpha \in \Omega_\alpha} \left\{ \lambda_+ \epsilon_+ + \lambda_- \epsilon_- + \widehat{p} \underset{\widehat{P}_+}{\mathbb{E}} \left[ \phi_{\mathbf{w}, \lambda_+, \alpha}(z) \right] + (1 - \widehat{p}) \underset{\widehat{P}_-}{\mathbb{E}} \left[ \phi_{\mathbf{w}, \lambda_-, \alpha}(z) \right] \right\} \Bigg]$$

$$\leq \sup_{\boldsymbol{\theta} \in \Theta} \Bigg[ \min_{(a,b) \in \Omega_{a,b}} \min_{\lambda_+, \lambda_- \geq 0} \max_{\alpha \in \Omega_\alpha} \left\{ \lambda_+ \epsilon_+ + \lambda_- \epsilon_- + p \underset{P_+}{\mathbb{E}} \left[ \phi_{\mathbf{w}, \lambda_+, \alpha}(z) \right] + (1 - p) \underset{P_-}{\mathbb{E}} \left[ \phi_{\mathbf{w}, \lambda_-, \alpha}(z) \right] \right.$$

$$- \min_{(a,b) \in \Omega_{a,b}} \min_{\lambda_+, \lambda_- \geq 0} \max_{\alpha \in \Omega_\alpha} \left\{ \lambda_+ \epsilon_+ + \lambda_- \epsilon_- + \widehat{p} \underset{\widehat{P}_+}{\mathbb{E}} \left[ \phi_{\mathbf{w}, \lambda_+, \alpha}(z) \right] + (1 - \widehat{p}) \underset{\widehat{P}_-}{\mathbb{E}} \left[ \phi_{\mathbf{w}, \lambda_-, \alpha}(z) \right] \right\} \Bigg]$$

By applying Lem.2 three times for $(a, b) \in \Omega_{a,b}$, $\min_{\lambda_+ \geq 0, \lambda_- \geq 0}$ and $\alpha \in \Omega_\alpha$, we have:

$$\sup_{\boldsymbol{\theta} \in \Theta} \Bigg[ DRAUC_{\epsilon_+, \epsilon_-}^{Da}(f_{\boldsymbol{\theta}}, P)$$

$$- \min_{(a,b) \in \Omega_{a,b}} \min_{\lambda_+, \lambda_- \geq 0} \max_{\alpha \in \Omega_\alpha} \left\{ \lambda_+ \epsilon_+ + \lambda_- \epsilon_- + \widehat{p} \cdot \underset{\widehat{P}_+}{\mathbb{E}} \left[ \phi_{\mathbf{w}, \lambda_+, \alpha}(z) \right] + (1 - \widehat{p}) \underset{\widehat{P}_-}{\mathbb{E}} \left[ \phi_{\mathbf{w}, \lambda_-, \alpha}(z) \right] \right\} \Bigg]$$

$$\leq \sup_{\boldsymbol{\theta} \in \Theta, (a,b) \in \Omega_{a,b}, \lambda_+ \geq 0, \lambda_- \geq 0} \Bigg[ p \cdot \underset{P_+}{\mathbb{E}} \left[ \phi_{\mathbf{w}, \lambda_+, \alpha}(z) \right] - \widehat{p} \cdot \underset{\widehat{P}_+}{\mathbb{E}} \left[ \phi_{\mathbf{w}, \lambda_+, \alpha}(z) \right]$$

$$+ (1 - p) \cdot \underset{P_-}{\mathbb{E}} \left[ \phi_{\mathbf{w}, \lambda_-, \alpha}(z) \right] - (1 - \widehat{p}) \cdot \underset{\widehat{P}_-}{\mathbb{E}} \left[ \phi_{\mathbf{w}, \lambda_-, \alpha}(z) \right] \Bigg]$$

$$\leq \sup_{\boldsymbol{\theta} \in \Theta, (a,b) \in \Omega_{a,b}, \lambda_+ \geq 0} \left[ p \cdot \underset{P_+}{\mathbb{E}} \left[ \phi_{\mathbf{w}, \lambda_+, \alpha}(z) \right] - \widehat{p} \cdot \underset{\widehat{P}_+}{\mathbb{E}} \left[ \phi_{\mathbf{w}, \lambda_+, \alpha}(z) \right] \right] +$$

$$\sup_{\boldsymbol{\theta} \in \Theta, (a,b) \in \Omega_{a,b}, \lambda_- \geq 0} \left[ (1 - p) \cdot \underset{P_-}{\mathbb{E}} \left[ \phi_{\mathbf{w}, \lambda_-, \alpha}(z) \right] - (1 - \widehat{p}) \cdot \underset{\widehat{P}_-}{\mathbb{E}} \left[ \phi_{\mathbf{w}, \lambda_-, \alpha}(z) \right] \right]$$

For the positive part, we have:

$$\sup_{\boldsymbol{\theta}\in\Theta,(a,b)\in\Omega_{a,b},\lambda_+\geq 0}\left[p\cdot\mathop{\mathbb{E}}_{P_+}[\phi_{\mathbf{w},\lambda_+,\alpha}(z)]-\widehat{p}\cdot\mathop{\mathbb{E}}_{\widehat{P}_+}[\phi_{\mathbf{w},\lambda_+,\alpha}(z)]\right]$$

$$\leq\sup_{\boldsymbol{\theta}\in\Theta,(a,b)\in\Omega_{a,b},\lambda_+\geq 0}\left[p\cdot\mathop{\mathbb{E}}_{P_+}[\phi_{\mathbf{w},\lambda_+,\alpha}(z)]-\widehat{p}\cdot\mathop{\mathbb{E}}_{P_+}[\phi_{\mathbf{w},\lambda_+,\alpha}(z)]+\widehat{p}\cdot\mathop{\mathbb{E}}_{P_+}[\phi_{\mathbf{w},\lambda_+,\alpha}(z)]-\widehat{p}\cdot\mathop{\mathbb{E}}_{\widehat{P}_+}[\phi_{\mathbf{w},\lambda_+,\alpha}(z)]\right]$$

$$\leq\sup_{\boldsymbol{\theta}\in\Theta,(a,b)\in\Omega_{a,b},\lambda_+\geq 0}\left[(p-\widehat{p})\cdot\mathop{\mathbb{E}}_{P_+}[\phi_{\mathbf{w},\lambda_+,\alpha}(z)]\right]+\widehat{p}\cdot\sup_{\boldsymbol{\theta}\in\Theta,(a,b)\in\Omega_{a,b},\lambda_+\geq 0}\left[\mathop{\mathbb{E}}_{P_+}[\phi_{\mathbf{w},\lambda_+,\alpha}(z)]-\mathop{\mathbb{E}}_{\widehat{P}_+}[\phi_{\mathbf{w},\lambda_+,\alpha}(z)]\right]$$

$$\leq C_\infty\cdot|p-\widehat{p}|+\widehat{p}\cdot\sup_{\boldsymbol{\theta}\in\Theta,(a,b)\in\Omega_{a,b},\lambda_+\geq 0}\left[\mathop{\mathbb{E}}_{P_+}[\phi_{\mathbf{w},\lambda_+,\alpha}(z)]-\mathop{\mathbb{E}}_{\widehat{P}_+}[\phi_{\mathbf{w},\lambda_+,\alpha}(z)]\right].$$

Similar, we have for the negative part:

$$\sup_{\boldsymbol{\theta}\in\Theta,(a,b)\in\Omega_{a,b},\lambda_-\geq 0}\left[(1-p)\cdot\mathop{\mathbb{E}}_{P_-}[\phi_{\mathbf{w},\lambda_-,\alpha}(z)]-(1-\widehat{p})\cdot\mathop{\mathbb{E}}_{\widehat{P}_-}[\phi_{\mathbf{w},\lambda_-,\alpha}(z)]\right]$$

$$\leq C_\infty\cdot|p-\widehat{p}|+(1-\widehat{p})\cdot\sup_{\boldsymbol{\theta}\in\Theta,(a,b)\in\Omega_{a,b},\lambda_-\geq 0}\left[\mathop{\mathbb{E}}_{P_-}[\phi_{\mathbf{w},\lambda_-,\alpha}(z)]-\mathop{\mathbb{E}}_{\widehat{P}_-}[\phi_{\mathbf{w},\lambda_-,\alpha}(z)]\right].$$

The result then follows directly.

$\square$

*Proof of Thm.4.* For the sake of simplicity, we denote:

$$\textbf{(a)}=\sup_{\boldsymbol{\theta}\in\Theta,(a,b)\in\Omega_{a,b},\lambda_+\geq 0}\left[\mathop{\mathbb{E}}_{P_+}[\phi_{\mathbf{w},\lambda_+,\alpha}(z)]-\mathop{\mathbb{E}}_{\widehat{P}_+}[\phi_{\mathbf{w},\lambda_+,\alpha}(z)]\right]$$

$$\textbf{(b)}=\sup_{\boldsymbol{\theta}\in\Theta,(a,b)\in\Omega_{a,b},\lambda_-\geq 0}\left[\mathop{\mathbb{E}}_{P_-}[\phi_{\mathbf{w},\lambda_-,\alpha}(z)]-\mathop{\mathbb{E}}_{\widehat{P}_-}[\phi_{\mathbf{w},\lambda_-,\alpha}(z)]\right]$$

$$\textbf{(c)}=|p-\widehat{p}|$$

From the Rademacher-complexity-based uniform convergence result, we have, with probability at least $1-\frac{\delta}{4}$:

$$\textbf{(a)}\leq 2\cdot\widehat{\mathfrak{R}}^+_{\widehat{P}_+}(\Theta)+C_+\cdot\sqrt{\frac{\log(8/\delta)}{2n_+}}$$

where $C_+$ is a universal constant.

Similarly, we have, with probability at least $1-\frac{\delta}{4}$:

$$\textbf{(b)}\leq 2\cdot\widehat{\mathfrak{R}}^-_{\widehat{P}_-}(\Theta)+C_-\cdot\sqrt{\frac{\log(8/\delta)}{2n_-}}$$

where $C_-$ is a universal constant. From the Chernoff bound, we have, with probability at least $1-\frac{\delta}{2}$:

$$\textbf{(c)}\leq\sqrt{\frac{\log(1/\delta)}{2n}}$$

Following the union bound and Lem.3, we have the following result holds for all $\boldsymbol{\theta}\in\Theta,(a,b)\in\Omega_{a,b},\alpha\in\Omega_\alpha,\lambda_+\geq 0,\lambda_-\geq 0$, the following holds with probability at least $1-\delta$:

$$DRAUC^{Da}_{\epsilon_+,\epsilon_-}(f_{\boldsymbol{\theta}},P)\leq\widehat{\mathcal{L}}+2\cdot\widehat{p}\cdot\widehat{\mathfrak{R}}^+_{\widehat{P}_+}(\Theta)+2\cdot(1-\widehat{p})\cdot\widehat{\mathfrak{R}}^-_{\widehat{P}_-}(\Theta)+$$

$$C_+\cdot\widehat{p}\cdot\sqrt{\frac{\log(8/\delta)}{2n_+}}+C_-\cdot(1-\widehat{p})\cdot\sqrt{\frac{\log(8/\delta)}{2n_-}}+$$

$$2\cdot C_\infty\cdot\sqrt{\frac{\log(1/\delta)}{2n}}$$

$\square$

# B Experiments

## B.1 Datasets

We first introduce the dataset used in the following section:

- **MNIST** [22]: The MNIST dataset comprises 60,000 images of digits, each with a resolution of 28x28, and includes 6,000 images for each digit from 0 to 9. This dataset is partitioned into a training set containing 50,000 images and a testing set with 10,000 images. We also allocate 10,000 images from the training set to create a validation set.

- **CIFAR-10/CIFAR-100** [18]: CIFAR-10/CIFAR-100 features 60,000 images, each having a resolution of 32x32x3, equally distributed across 10/100 classes and containing 6,000/600 images per class. The dataset is separated into a training set of 50,000 images and a testing set of 10,000 images. In addition, we extract 10,000 images from the training set to form a validation set.

- **Tiny-ImageNet** [21]: The Tiny-ImageNet dataset comprises 110,000 images in 200 classes, including 100,000 training examples and 10,000 testing examples. We further split off a validation set containing 20,000 examples from the training set. We find that generating a binary Tiny-ImageNet-200 by assigning the first half of the classes as positive and the rest as negative makes this dataset too challenging to learn. The methods struggle to learn good features and reach a testing AUC no larger than 0.6. As a result, we assign the binary version of Tiny-ImageNet by utilizing the hyper-class information. As detailed, we construct three subsets as follows:
  - Tiny-ImageNet-200-Dogs: Classes [11, 39, 78, 135, 182, 194] are assigned as positive, with the remainder designated as negative.
  - Tiny-ImageNet-200-Birds: Classes [35, 41, 67, 115] are assigned as positive, with the remainder designated as negative.
  - Tiny-ImageNet-200-Vehicles: Classes [15, 64, 69, 75, 90, 108, 114, 117, 147, 152, 157, 163] are assigned as positive, with the remainder designated as negative.

- **MNIST-C** [30]: MNIST-C is a corrupted variant of the original MNIST testing set, consisting of 160,000 testing examples generated through 16 distinct perturbation techniques (including identity transform) tailored for handwritten digits.

- **CIFAR-10-C/CIFAR-100-C** [13]: The CIFAR-10-C/CIFAR-100-C datasets are corrupted versions of the original CIFAR-10/CIFAR-100 testing sets, encompassing 950,000 images obtained by applying five intensity levels of 19 different corruption types, such as noises, blurs, and transformations. We analyze the average performance for each corruption level.

- **Tiny-ImageNet-C** [14]: The Tiny-ImageNet-C is the corrupted version of Tiny-ImageNet. 5 levels of 15 different corruptions including brightness, compression and blurs are applied to 10,000 images to generate 950,000 testing images.

## B.2 Dataset Constructions

We construct our binary long-tailed dataset following a manner similar to [48]. First, we construct a binary version of the dataset by assigning the former half of the classes as positive and the latter half as negative. Then, utilizing the imbalance ratio, i.e. {0.01, 0.05, 0.1, 0.2} in our configuration, we randomly eliminate a portion of positive samples to create the long-tailed version. For instance, to produce CIFAR10-LT with an imbalance ratio of 0.1, we designate classes 0-4 as positive and classes 5-9 as negative. Subsequently, we randomly remove $\approx 89\%$ of the training samples to achieve the desired long-tailed dataset.

## B.3 Competitors

To confirm the robustness of our proposed method in imbalanced scenarios, we compare it with the following competitors, each corresponding to one row in Table 1:

- **Baseline**: Cross-entropy loss (**CE**).
- **Typical methods for long-tailed problems**:

- **FocalLoss** [24]: A classical reweighting method for long-tailed problems.
- **AUCMLoss** [51]: An instance-wise binary AUC optimization technique.
- **AUCDRO** [56, 53]: A method that integrates DRO technique with partial AUC optimization.

- **DRO methods**:
  - **ADVShift** [55]: A DRO method addressing label shift.
  - **WDRO** [20]: A Wasserstein DRO technique incorporating local perturbations.
  - **DROLT** [37]: A loss function designed to learn low-variance representations.
  - **GLOT** [34]: A regularization method based on optimal transport distributional robustness.

- **Our approach**:
  - Our Algorithm 1 (**DRAUC-Df**).
  - Our Algorithm 2 (**DRAUC-Da**).

## B.4 Implementation Details

We conducted all experiments on a `Ubuntu 20.04.5` server, equipped with an `Intel(R) Xeon(R) Gold 6230R CPU` and an `RTX 3090 GPU`. All codes were implemented using `PyTorch` (v-1.8.2) [33], `TorchVision` (v-0.9.2), and `Numpy` (v-1.21.4) under a `Python` 3.8 and `CUDA` 11.1 environment.

For our models, we selected ResNet20 [12], ResNet32, and Small CNN as the backbone architectures. The models were trained for 100 epochs across all datasets. On the CIFAR10, CIFAR100 and Tiny-ImageNet datasets, we applied random cropping with padding and random horizontal flipping as data augmentation techniques. However, for the MNIST dataset, we refrained from applying any data augmentation because the horizontal flip could alter the semantic meaning of the digits. For all experiments, we set the weight decay to $5 \times 10^{-4}$ and the batch size to 128. During training, we utilized a sampler to ensure that at least one positive example was included in each batch.

## B.5 Choices of Hyperparameters

**Initial Learning Rate and Learning Rate Scheduler.** We selected the initial learning rate from the set $0.01, 0.05, 0.1, 0.2$. In the majority of cases that are not extremely imbalanced, $lr = 0.1$ is a favorable choice. However, in situations where the dataset is extremely imbalanced, careful tuning of the initial learning rate is necessary. We chose the learning rate scheduler from a step scheduler, which decays the learning rate by $10\times$ at the 50-th and 75-th epochs, and the Cosine Annealing scheduler.

**Robust Diameter $\epsilon$.** We selected $\epsilon$ from the set $\{8/255, 32/255, 64/255, 128/255\}$, considering $l_2$ distance. A sensitivity analysis regarding $\epsilon$ is presented in Section 5.2.2. For distribution-aware DRAUC, we chose $\epsilon_+$ and $\epsilon_-$ using the following approach: Given an overall diameter $\epsilon$ and a tunable parameter $k \in \{0.5, 0.8, 1, 1.2, 1.5\}$, we set $\epsilon_+ = k\epsilon$ and $\epsilon_- = (1 - k\widehat{p})\epsilon/(1 - \widehat{p})$.

**Learning Rates for Tunable Parameters.** We selected $\eta_\alpha = \eta_w = lr$ and $\eta_z = 15/255$, which aligns with the standard settings in Adversarial Training [36]. For $\eta_\lambda$, we chose from the set $\{0.01, 0.02, 0.1, 0.2\}$. A sensitivity analysis regarding $\eta_\lambda$ is detailed in Section 5.2.2.

**Initialization of Tunable Parameters.** For initialization, we set $\lambda_0 = 1$, $a^0 = 0$, $b^0 = 0$, $\alpha^0 = 0$, and PGD steps $K = 10$.

## B.6 Additional Empirical Results

In Table 3, we display the overall performance metrics for CIFAR100-C and CIFAR100-LT, while Table 4 illustrates the performance for MNIST-C and MNIST-LT. Additionally, the overall performance under varying perturbation levels is presented in Figures 3-5. We have not included the results for MNIST-C due to the original MNIST-C [30] only providing a single perturbation level. These comprehensive results facilitate several observations, as detailed in Section 5.2.1:

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

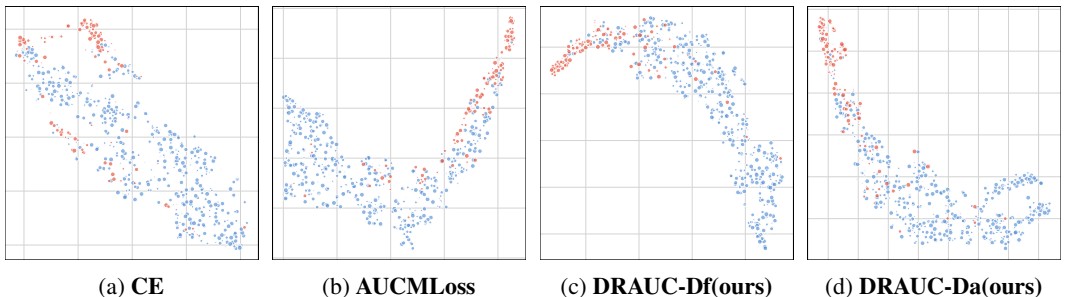

(a) **CE**  (b) **AUCMLoss**  (c) **DRAUC-Df(ours)**  (d) **DRAUC-Da(ours)**

Figure 8: t-SNE plots of model embeddings on CIFAR10-C.

- Our methods consistently outperform all competitors on the corrupted datasets, across varying imbalance ratios and model architectures, confirming the effectiveness of our proposed method.
- Our methods achieve superior performance under stronger perturbations, thereby substantiating that our proposed methods enhance model robustness. This inference can be considered an ablation result.
- In most cases, distribution-aware contributes to improving model robustness.

### B.7 Visualizations

In this section, we provide more visualization results.

**t-SNE Plots.** We display the t-SNE plots for CIFAR10-C, CIFAR100-C and MNIST-C in Figures 8, 9 and 10. As evident from the plots, the embeddings on CIFAR100-C are more challenging to separate than those on CIFAR10-C and MNIST-C. This outcome primarily due to two factors: **a)** The number of patterns in CIFAR100 exceeds those in CIFAR10 and MNIST. **b)** When we create our binary version datasets, we designate the first half of classes as positive and the remaining half as negative. Consequently, the positive class of CIFAR100 comprises 50 original classes, making it more complex to learn, and we should anticipate a larger inner-class variance of its embeddings.

However, as demonstrated by the plots, our proposed method offers a more separable embedding space compared to the baselines.

**An Interpretation of DRAUC's Improvement on Model Generalization for Corrupted Data.** We provide several examples generated by our method in Figure 11. The results demonstrate that even without prior knowledge of the corruptions in the testing distribution, our DRAUC method generates adversarial examples closely resembling the test corruptions, thereby enhancing the model's resistance to them.

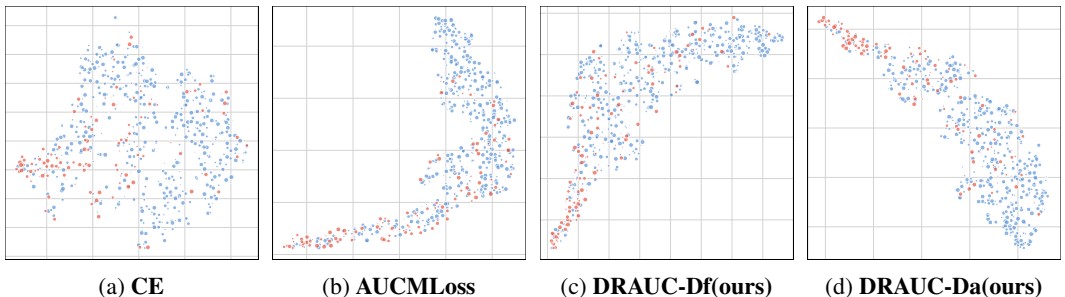

|         |            |                  |                  |
|:-------:|:----------:|:----------------:|:----------------:|
| (a) **CE** | (b) **AUCMLoss** | (c) **DRAUC-Df(ours)** | (d) **DRAUC-Da(ours)** |

Figure 9: t-SNE plots of model embeddings on CIFAR100-C.

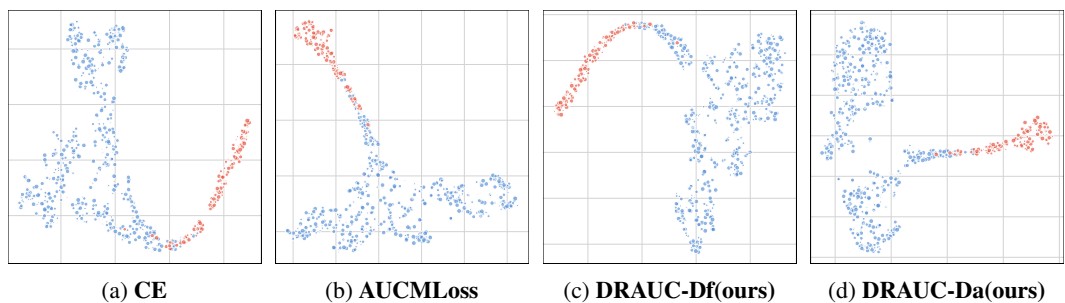

|         |            |                  |                  |
|:-------:|:----------:|:----------------:|:----------------:|
| (a) **CE** | (b) **AUCMLoss** | (c) **DRAUC-Df(ours)** | (d) **DRAUC-Da(ours)** |

Figure 10: t-SNE plots of model embeddings on MNIST-C.

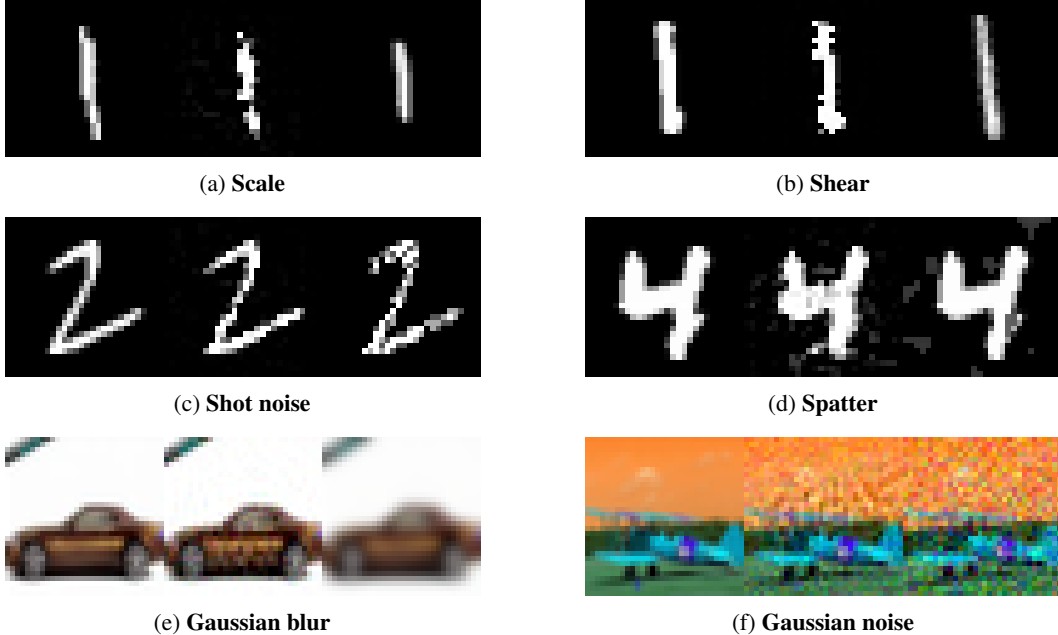

(a) **Scale**

(b) **Shear**

(c) **Shot noise**

(d) **Spatter**

(e) **Gaussian blur**

(f) **Gaussian noise**

Figure 11: **Visualizations of adversarial examples generated by DRAUC-Df.** Each group of images represent original image, adversarial image generated by DRAUC-Df and the corrupted image in the testing set, respectively.