# OpenReview forum: "DRAUC: An Instance-wise Distributionally Robust AUC Optimization Framework"
_NeurIPS.cc/2023/Conference — NeurIPS 2023 poster_

### Official Review · Reviewer_uBXL · 2023-07-07

**Soundness:** 3 good
**Presentation:** 3 good
**Contribution:** 3 good
**Rating:** 6
**Confidence:** 1

**Summary:**

This paper considers AUC optimization in the distribution shift setting, which is of great value to the machine learning community. The author formulates the problem as a min-max-min-max problem and relaxes it as an upper bound minimization problem. The experimental results also confirm the effectiveness of this proposal.

**Strengths:**

The theoretical results seem reasonable. However, as I am not an expert on distributionally robust optimization, I cannot justify the theoretical significance of this paper.

**Weaknesses:**

Table 1 shows the results of CIFAR10-LT, but there is no description of the experimental settings in the context. The proposed method performs poorly on CIFAR10-LT, which seems to contradict the claim of the article. How can this phenomenon be explained?

Miscellaneous:

The colorful curve in Figure 1 may not be friendly for black-and-white print versions. Providing a legend directly may be a better choice.

The definition of $b^*$ in Eq (5) seems to be incorrect.

**Questions:**

Is there more discussion on the results of CIFAR10-LT, why is the performance of the proposal similar to AUCMLoss, and sometimes even weaker than AUCMLoss?

---
Thanks for the detailed clarifications, which have addressed my concerns. I would like to keep my score.

**Limitations:**

The authors do not provide a discussion about the limitations and potential negative societal impact.

---

> ### Author Rebuttal · Authors · 2023-08-09
>
> Thank you for your detailed comments! Our response is as follows:
>
> >**Q1:** Table 1 shows the results of CIFAR10-LT, but there is no description of the experimental settings in the context. The proposed method performs poorly on CIFAR10-LT, which seems to contradict the claim of the article. How can this phenomenon be explained?
>
> **A1:** Thank you for your valuable question! We appreciate your feedback regarding the ambiguity in our paper and apologize for any confusion. To construct the binary, long-tailed version of the dataset, we followed the methodology proposed in [25]. The first half of the classes are assigned as positive, while the remainder are designated as negative. Subsequently, some examples are excluded to establish a desired imbalance ratio. We plan to relocate this section from the Appendix to the main body of the paper for better comprehension.
>
> Responding to your query, we would like to address it from two perspectives. First, we do not directly optimize performance on the clean CIFAR10-LT. Our primary focus is to enhance performance on the local worst-case distribution surrounding the original, uncorrupted distribution. In our experiments, this corresponds to the **corrupted versions** of datasets.
>
> Here, we directly list the empirical result on corrupted datasets of our proposed methods and the baseline (AUCMLoss), so you can check our progress directly.
>
> |Datasets|Models|Methods|0.01|0.05|0.10|0.20|
> |:-:|:-:|:-:|:-:|:-:|:-:|:-:|
> |CIFAR10|resnet20|AUCMLoss|63.93|76.77|81.75|85.26|
> |CIFAR10|resnet20|DRAUC-Df|65.3(+1.37)|79.87(+3.1)|84.33(+2.58)|89.0(+3.74)|
> |CIFAR10|resnet20|DRAUC-Da|65.69(+1.76)|80.05(+3.28)|86.17(+4.42)|89.7(+4.44)|
> |CIFAR10|resnet32|AUCMLoss|64.00|76.98|81.87|85.66|
> |CIFAR10|resnet32|DRAUC-Df|64.66(+0.66)|80.25(+3.27)|85.88(+4.01)|90.63(+4.97)|
> |CIFAR10|resnet32|DRAUC-Da|65.38(+1.38)|80.84(+3.86)|86.28(+4.41)|89.44(+3.78)|
> |CIFAR100|resnet20|AUCMLoss|55.70|61.91|65.73|69.58|
> |CIFAR100|resnet20|DRAUC|57.27(+1.57)|62.43(+0.52)|66.25(+0.52)|71.34(+1.76)|
> |CIFAR100|resnet20|CDRAUC|57.5(+1.8)|62.28(+0.37)|66.11(+0.38)|71.06(+1.48)|
> |CIFAR100|resnet32|AUCMLoss|56.19|61.87|63.64|69.81|
> |CIFAR100|resnet32|DRAUC|57.04(+0.85)|61.9(+0.03)|65.96(+2.32)|71.22(+1.41)|
> |CIFAR100|resnet32|CDRAUC|57.38(+1.19)|62.12(+0.25)|66.13(+2.49)|71.31(+1.5)|
> |MNIST|small_cnn|AUCMLoss|95.52|98.16|98.04|98.60|
> |MNIST|small_cnn|DRAUC|96.06(+0.54)|98.38(+0.22)|98.53(+0.49)|98.84(+0.24)|
> |MNIST|small_cnn|CDRAUC|96.35(+0.83)|98.04(+-0.12)|98.56(+0.52)|98.92(+0.32)|
> |MNIST|resnet20|AUCMLoss|89.09|97.82|96.26|97.74|
> |MNIST|resnet20|DRAUC|94.72(+5.63)|98.21(+0.39)|98.32(+2.06)|98.67(+0.93)|
> |MNIST|resnet20|CDRAUC|94.52(+5.43)|98.01(+0.19)|98.53(+2.27)|98.79(+1.05)|
>
> Here are the experiment results we add in the rebuttal phase:
> |Datasets|Models|Methods|Dog|Bird|Vehicles|
> |:-:|:-:|:-:|:-:|:-:|:-:|
> |TINYIMAGENET|resnet20|AUCMLoss|77.35|85.98|82.37|
> |TINYIMAGENET|resnet20|DRAUC-Df|84.11(+6.76)|87.3(+1.32)|88.67(+6.3)|
> |TINYIMAGENET|resnet20|DRAUC-Da|83.96(+6.61)|87.61(+1.63)|89.06(+6.69)|
> |TINYIMAGENET|resnet32|AUCMLoss|77.25|85.20|81.12|
> |TINYIMAGENET|resnet32|DRAUC-Df|85.79(+8.54)|88.0(+2.8)|88.32(+7.2)|
> |TINYIMAGENET|resnet32|DRAUC-Da|84.56(+7.31)|87.6(+2.4)|88.46(+7.34)|
>
> |Datasets|Models|Methods|Results|
> |:-:|:-:|:-:|:-:|
> |MELANOMA|efficientnetb0|AUCMLoss|73.95|
> |MELANOMA|efficientnetb0|DRAUC-Df|77.82(+3.87)|
> |MELANOMA|efficientnetb0|DRAUC-Da|78.54(+4.59)|
> |MELANOMA|densenet121|AUCMLoss|75.67|
> |MELANOMA|densenet121|DRAUC-Df|77.89(+2.22)|
> |MELANOMA|densenet121|DRAUC-Da|76.82(+1.15)|
>
>
>
> Secondly, due to the phenomenon of the **trade-off between clean accuracy and robust accuracy** in adversarial training [a], in many cases, achieving optimal performance on both clean and adversarial examples simultaneously is unattainable. This phenomenon can explain the performance drop in the clean data.
>
>
> ---
>
> >**Q2:** The colorful curve in Figure 1 may not be friendly for black-and-white print versions. Providing a legend directly may be a better choice.
>
> **A2:** Thank you for your kind suggestion. We will add a legend.
>
> ---
>
> >**Q3:** The definition of $b^*$ in Eq (5) seems to be incorrect.
>
> **A3:** Thanks for the detailed review, the corrected version is $b^* = \mathbb E_{P^-}[f_{{\theta}}({x}^-)]$.
>
> ---
>
> >**Q4:** Is there more discussion on the results of CIFAR10-LT, why is the performance of the proposal similar to AUCMLoss, and sometimes even weaker than AUCMLoss?
>
> **A4:** Thanks again for your valuable question! As discussed in **A1**, we attribute this phenomenon to the inherent trade-off between clean AUC and robust AUC. AUCMLoss can be perceived as an optimal classifier for clean data. However, it is vulnerable to distributional attacks. In contrast, our DRAUC might not perform as well as AUCMLoss on clean data, but it demonstrates superior performance on robust data.
>
> ---
>
> [a]Zhang, Hongyang, et al. "Theoretically principled trade-off between robustness and accuracy." International conference on machine learning. PMLR, 2019.

---

> > ### Comment · Reviewer_uBXL · 2023-08-16
> >
> > Thank you for the clarifications and additional supporting results.

---

> > > ### Author Response · Authors · 2023-08-16
> > >
> > > Thank you for your comment! Please feel free to ask if there is any confusion, and we are glad to address any concern you may have.

---

### Official Review · Reviewer_u8fo · 2023-07-08

**Soundness:** 3 good
**Presentation:** 3 good
**Contribution:** 3 good
**Rating:** 5
**Confidence:** 1

**Summary:**

Area under the ROC curve (AUC) has been widely used for imbalanced classification, but optimizing the metric requires the assumption that training and test data come from the same distribution, which is not realistic. The authors tackle to explore distributionally robust optimization under the context of AUC optimization. They first integrate DRO and AUC and reformulate it to a tractable optimization problem using surrogate loss functions. Finally, they proposed distribution-free DRAUC and distribution-aware DRAUC optimization frameworks. The proposed frameworks are validated theoretically and empirically via experiments on multiple datasets.

**Strengths:**

- The authors propose DRAUC-df by combining DRO and AUC optimization in a tractable way and further extend it to DRAUC-da by considering perturbations on the positive and negative distributions separately.
- Two proposed frameworks are validated on several standard datasets, which achieved meaningful performance improvement comparing to the baseline (AUCML).

**Weaknesses:**

- Experiments are only conducted on small-scale datasets, like as CIFAR or MNIST with ResNet and CNN architectures. It may need to validate on more realistic large-scale datasets, like as AUCML on large-scale medical image datasets with more recent network architectures (EfficientNet, DenseNet).
- This paper addresses the robustness under worst local distribution from DRO perspective, but it restricts to binary classification problem (potentially due to the use of AUC metric) and validated only on binary version of datasets, so I have reservations about the suitability of this problem setting.


**Questions:**

- In L97, why the Wasserstein constraint in the optimization algorithms can be removed with Theorem 2?
- Typos in the definition of \ell_{0,1} and Eq. (2)

**Limitations:**

Limitations and potential societal impacts are not addressed.

---

> ### Author Rebuttal · Authors · 2023-08-09
>
> Thank you for your constructive reviews! To address your concern, we conduct several more experiments on larger datasets, and expand our method toward multi-class settings.
>
> >**Q1:** Experiments are only conducted on small-scale datasets. It may need to validate on more realistic large-scale datasets, like as AUCML on large-scale medical image datasets with more recent network architectures (EfficientNet, DenseNet).
>
> **R1:** Thank you for your valuable question. To address your concerns, we conduct additional experiments on larger datasets, including **Tiny-ImageNet-200** and a large-scale medical image dataset **Melanoma**, utilizing **EfficientNet** and **DenseNet**. The outcomes validate our results' applicability to extensive datasets and modern network architectures.
> Here, we use **bold** to mark the highest scores, and the results with '-C' represent the testing performance in corrupted versions of datasets. **Due to the limitation of reply, please refer to Table 1 and 2 in the pdf of the global response for full results!**
>
> **Empirical results on Melanoma：**
> |Methods|Mela-C(Enet-b0)|Mela-C(Dense-121)|Mela(Enet-b0)|Mela(Dense-121)|
> |:-:|:-:|:-:|:-:|:-:|
> |CE|69.65|72.05|83.27|83.37|
> |AUCMLoss|73.95|75.67|85.98|84.33|
> |FocalLoss|65.64|72.97|78.81|84.64|
> |ADVShift|70.98|71.91|80.34|81.60|
> |WDRO|75.88|74.71|86.29|**85.79**|
> |DROLT|71.50|76.04|82.89|83.30|
> |GLOT|71.52|69.83|83.95|81.59|
> |AUCDRO|74.57|76.09|**86.77**|85.51|
> |DRAUC-Df|77.82|**77.89**|85.06|81.44|
> |DRAUC-Da|**78.54**|76.82|82.98|81.61|
>
> ---
>
> >**Q2:** This paper addresses the robustness under worst local distribution from DRO perspective, but it restricts to binary classification problem (potentially due to the use of AUC metric) and validated only on binary version of datasets, so I have reservations about the suitability of this problem setting.
>
> **R2: Thanks for this heuristic question! We take a trial on expanding our method toward multi-class setting, and empirically prove that our proposed method also works well in multi-class cases.**
> AUC is often used in binary classification problems. However, we can generalize our method to accommodate multi-class settings by interpreting a multi-class classification problem as a series of binary decision-making processes. First, we define multi-class AUC:
>
> $AUCM(f\_{{\theta}}) = \frac{1}{N\_C}\sum\_{i=1}^{N\_C}AUC\_i(f\_{{\theta}}) = \frac{1}{N\_C}\sum\_{i=1}^{N\_C} \mathbb E\_{\widehat P\_i, {\widehat P\_i'}}[\ell(f\_{{\theta}}(x\_i) -f\_{{\theta}}(x\_i'))]$
>
> where $N\_C$ is the number of classes, $AUC\_i$ denotes the AUC score when assigning $i^{th}$ class as positive, and the rest classes as negative, $\widehat P\_i$ denotes the example distribution of $i^{th}$ class and $\widehat P\_i'$ denotes the distribution or the rest classes. Consistent with the notation in our paper, we use $\widehat Q$ to describe the perturbed distribution. Following the idea of optimizing the multi-class AUC under local worst distribution, we now present the objective of Multi-class DRAUC(MDRAUC)
>
>
> $\min\_{{\theta}}\max\_{\widehat Q:\mathcal W\_c(\widehat Q,\widehat P)\le \epsilon}  ~\mathbb E\_{\widehat Q}\left[\frac 1{N\_C}\sum\_{i=1}^{N\_C}AUC\_i(f\_{{\theta}})\right]$
>
> Our formulation of MDRAUC-Df is as follows:
>
> ${(MDf\star)}\quad\min\_{{{\theta}},\vec a,\vec b} \min\_{\lambda > 0}\max\_{\vec \alpha} \left\\{\lambda\epsilon + \mathbb E\_{\widehat{ P}}[\widehat \varphi\_{{{\theta}}, \vec a, \vec b, \vec \alpha, \lambda}(z)] \right\\}$
>
> where
>
> $\widehat \varphi\_{{{\theta}}, \vec a, \vec b, \vec \alpha, \lambda}(z) = \max\_{z' \in \mathcal Z}\left[\frac 1{N\_C}\sum\_{i=1}^{N\_C}g\_i(a\_i,b\_i,\alpha\_i,{{\theta}},z) - \lambda c(z,z')\right].$
>
> Limited by the length of reply, we place the **detailed derivation, experiment setting, and full results (Table 3 in the pdf file)** in the **global response**. We kindly refer you to get more details from it.
> We further validate the effectiveness of the above multi-class DRAUC algorithm on CIFAR10-LT, and the experiment results are as follows:
>
> **Empirical results of ResNet20 on Multi-class-CIFAR10-LT**
> Methods|0.01-C|0.05-C|0.10-C|0.20-C|0.01|0.02|0.10|0.20|
> |:-:|:-:|:-:|:-:|:-:|:-:|:-:|:-:|:-:|
> |CE|89.48|90.95|91.58|92.07|95.95|97.98|**98.58**|98.92|
> |AUCMLoss|86.77|88.66|90.88|91.29|95.31|97.69|98.21|98.56|
> |FocalLoss|89.82|91.27|91.97|91.67|**96.42**|**98.15**|98.56|98.95|
> |ADVShift|70.42|72.48|76.54|74.48|72.15|80.87|82.47|83.96|
> |WDRO|90.01|91.30|91.22|91.68|96.35|98.04|98.57|**98.96**|
> |DROLT|89.01|88.72|91.18|90.42|95.44|97.19|98.20|98.52|
> |GLOT|90.44|91.88|91.63|92.49|96.29|97.85|98.57|98.88|
> |DRAUC|**91.28**|**93.15**|**94.98**|**95.74**|94.83|96.47|97.87|98.12|
> |CDRAUC|90.92|92.81|94.20|94.75|95.12|97.06|98.26|98.63|
>
> Our methods outperform all competitors in the corrupted dataset in a multi-class setting. We acknowledge that some previous works have studied advanced algorithms of multi-class AUC learning. However, current derivations and empirical results are enough to prove that our method is capable in multi-class settings, and we would like to leave further explorations of multi-class DRAUC as future work.
>
> ---
>
> >**Q3:** In L97, why the Wasserstein constraint in the optimization algorithms can be removed with Theorem 2?
>
> **R3:** Thanks for your valuable question! The Wasserstein constraint is not removed in Theorem 2; it is reformulated using the strong duality of Wasserstein DRO. Detailed proof can be found in [a].
>
> ---
>
> >**Q4:** Typos in the definition of $\ell\_{0,1}$ and Eq. (2)
>
> **R4:** Thank you for your detailed review. We will adjust our Eq. (1) to ensure consistency with the definition of $\ell\_{0,1}$, as follows:
>
> $AUC(f\_{{\theta}}) = {\mathbb E\_{P\_+,P\_+}\left[{\ell\_{0,1}(f\_{{\theta}}({{x}^+}) - f\_{{\theta}}({{x}^-}))}\right]}$
>
> ---
>
> [a]Gao, Rui, and Anton Kleywegt. "Distributionally robust stochastic optimization with Wasserstein distance." Mathematics of Operations Research 48.2 (2023): 603-655.

---

> > ### Comment · Area_Chair_D8vA · 2023-08-18
> >
> > Thank you for the rebuttal!

---

> > > ### Author Response · Authors · 2023-08-20
> > >
> > > Thank you for your comment!

---

> ### Author Response · Authors · 2023-08-20
>
> Dear reviewer u8fo,
>
> We would like to kindly inquire if our rebuttal adequately addresses your concerns. If you have further questions, please feel free to ask, and we will be glad to answer.

---

### Official Review · Reviewer_9FMm · 2023-07-09

**Soundness:** 3 good
**Presentation:** 3 good
**Contribution:** 3 good
**Rating:** 5
**Confidence:** 3

**Summary:**

The paper proposes an end-to-end framework to optimize the distributional robust AUC.
The authors address several key challenges of this task and robust generalization can be achieved with theoretical guarantee.
Experiments on several corrupted long-tailed benchmarks demonstrate the superiority of the proposed method.

**Strengths:**

1. The paper addresses several key challenges to integrating AUC optimization with DRO, including computational infeasibility, intractable solution, and label bias. The proposed method is well-motivated and technically sound. And the generalization risk is bounded with theoretical guarantee.

2. Experiments on several corrupted long-tailed benchmarks demonstrate the superiority of the proposed method.

3. The paper is well-written and clearly organized. Sufficient preliminaries on AUC and DRO are provided.


**Weaknesses:**

1. The proposed method is only evaluated on small-scale datasets such as MNIST and CIFAR-10/100. It remains unclear whether the proposed method can be scaled to large datasets.

2. It is well known that DRO suffers from the problem of yielding overly-pessimistic models with low confidence [1]. This weakness might be inherited in the proposed method as Wasserstein DRO is adopted.

[1] Hu, Weihua, et al. "Does distributionally robust supervised learning give robust classifiers?." International Conference on Machine Learning. PMLR, 2018.

**Questions:**

Please check the weakness.

**Limitations:**

The limitation of the proposed method may lie in scalability and the issue of overly-pessimistic models.

---

> ### Author Rebuttal · Authors · 2023-08-09
>
> Thank you for your constructive concerns! We would like to clarify the following questions:
>
> >**Q1:** The proposed method is only evaluated on small-scale datasets such as MNIST and CIFAR-10/100. It remains unclear whether the proposed method can be scaled to large datasets.
>
> **R1:** Thank you for your constructive question! To address your concern, we carried out several experiments on large-scale datasets, including **Melanoma** and **Tiny-ImageNet-200** (due to the limitation of length, kindly refer to the global response for the experiment settings). Additionally, to validate the adaptability of our proposed method across various network architectures, we incorporate the new backbones **EfficientNet** and **DenseNet**. Our results surpass all the competitors on large-scale datasets and network architectures, proving the effectiveness of our proposed method.
> For clarity in our presentation, we employ **bold** for the highest scores. Results appended with '-C' depict testing performance on corrupted dataset versions. You may also refer to the global response for experiment settings and better represented results(**Table 1 and 2 in the pdf of global response**).
>
> **Empirical results on Melanoma:**
> |Methods|Mela-C(Enet-b0)|Mela-C(Dense-121)|Mela(Enet-b0)|Mela(Dense-121)|
> |:-:|:-:|:-:|:-:|:-:|
> |CE|69.65|72.05|83.27|83.37|
> |AUCMLoss|73.95|75.67|85.98|84.33|
> |FocalLoss|65.64|72.97|78.81|84.64|
> |ADVShift|70.98|71.91|80.34|81.60|
> |WDRO|75.88|74.71|86.29|**85.79**|
> |DROLT|71.50|76.04|82.89|83.30|
> |GLOT|71.52|69.83|83.95|81.59|
> |AUCDRO|74.57|76.09|**86.77**|85.51|
> |DRAUC-Df|77.82|**77.89**|85.06|81.44|
> |DRAUC-Da|**78.54**|76.82|82.98|81.61|
>
> **Empirical results on Tiny-ImageNet-200(splited by hyper-class):**
> |Models|Methods|Dog-C|Bird-C|Vehicle-C|Dog|Bird|Vehicle|
> |:-:|:-:|:-:|:-:|:-:|:-:|:-:|:-:|
> |resnet20|CE|78.46|85.19|87.53|93.72|94.49|97.72|
> |resnet20|AUCMLoss|77.35|85.98|82.37|93.35|94.11|97.34|
> |resnet20|FocalLoss|78.34|81.48|86.55|93.25|92.87|97.66|
> |resnet20|ADVShift|81.20|80.94|86.65|93.70|93.53|97.66|
> |resnet20|WDRO|82.20|85.23|85.92|94.46|95.50|**98.19**|
> |resnet20|DROLT|80.44|86.91|86.76|93.89|**96.40**|97.86|
> |resnet20|GLOT|81.96|85.89|86.80|**94.67**|96.14|98.05|
> |resnet20|DROAUC|75.97|83.26|79.46|92.58|93.04|96.29|
> |resnet20|DRAUC-Df|**84.11**|87.30|88.67|93.39|95.58|97.50|
> |resnet20|DRAUC-Da|83.96|**87.61**|**89.06**|93.76|95.94|97.25|
> |resnet32|CE|82.55|84.64|86.26|94.31|94.49|97.76|
> |resnet32|AUCMLoss|77.25|85.20|81.12|93.19|95.19|97.57|
> |resnet32|FocalLoss|77.96|79.80|85.33|93.41|92.85|97.78|
> |resnet32|ADVShift|84.30|84.56|86.43|92.92|94.71|97.59|
> |resnet32|WDRO|80.08|85.58|86.94|94.39|95.51|97.67|
> |resnet32|DROLT|79.25|85.75|86.79|91.68|**96.06**|97.82|
> |resnet32|GLOT|81.70|83.09|88.24|94.08|95.16|**97.92**|
> |resnet32|DROAUC|78.21|80.55|85.26|91.56|93.15|96.33|
> |resnet32|DRAUC-Df|**85.79**|**88.00**|88.32|**94.43**|95.29|97.37|
> |resnet32|DRAUC-Da|84.56|87.60|**88.46**|94.03|95.96|97.65|
>
> ---
>
> >**Q2:** It is well known that DRO suffers from the problem of yielding overly-pessimistic models with low confidence [a]. This weakness might be inherited in the proposed method as Wasserstein DRO is adopted.
> [a] Hu, Weihua, et al. "Does distributionally robust supervised learning give robust classifiers?." International Conference on Machine Learning. PMLR, 2018.
>
> **R2: Thank you for your valuable question! We believe such an overly pessimistic problem will not appear in our setting for the following reasons:**
>
> While both methodologies employ DRO techniques to augment model robustness, the distinction between approaches leveraging $f$-divergence and those using Wasserstein DRO is essential. Specifically, the DRO based on $f$-divergence considers generating a local most adversarial reweight to maximize the training loss, represented as:
>
> $ \min\_{{\theta}}\sup\_{r\in\mathcal U\_f} \mathbb E\_{P}[r(z)\ell(g\_{{\theta}},z)] $
>
> $\mathcal U\_f \equiv \\{r(z) |\mathbb E\_{P}[f(r(z))]\le\epsilon, \mathbb E\_{P}[r(z)]=1, r(z)\ge 0 ,\forall z \in \mathcal Z \\}$
>
> In contrast, the Wasserstein DRO is designed to produce a local adversarial perturbation on the input to maximize the training loss:
>
> $\min\_{{\theta}}\max\_{\widehat Q:\mathcal W\_c(\widehat Q,\widehat P)\le \epsilon}  ~\mathbb E\_{\widehat Q}\left[\ell(g\_{{\theta}},z)\right]$
>
> For a clearer comparison, akin to the approach in [a], we first present the empirical approximated formulation for both $f$-divergence DRO and Wasserstein DRO as:
>
> ${(ERM)}\quad\min\_{{\theta}}\frac 1 N\sum\_{i=1}^N \ell(g\_{{\theta}},z)$
>
> ${(f-divergence \ DRO)}\quad\min\_{{\theta}}\sup\_{r\in\widehat{\mathcal U}\_f} \frac 1 N {\sum\_{i=1}^N}r\_i\cdot\ell(g\_{{\theta}},z)$
>
> $ {(Wasserstein~DRO)}\quad \min\_{{\theta}}\sup\_{z'\in\widehat{Q}} \frac 1 N\sum\_{i=1}^N \ell(g\_{{\theta}},z')$
>
> where
>
> $\widehat{\mathcal U}\_f = \left\\{ r |\frac 1 N\sum\_{i=1}\^N f(r_i) \le \epsilon \frac 1 N\sum\_{i=1}\^N r\_i = 1,  r \ge 0 \right\\},\widehat Q  =\\{Q:\mathcal W\_c(Q,\widehat P)\le \epsilon\\}$
>
> Being overly pessimistic implies that a model trained by $f$-divergence DRO may end up at a stationary point of the original ERM. This gives us an intuition that the adversarial reweight may be useless if structural assumptions are not introduced. As our approach does not engage in example reweighting, we avoid this overly pessimistic issue. Additionally, from an empirical perspective, we compare our results with AUCMLoss, which can be considered the ERM version of our method. The experimental results prove the enhanced robustness of our approach, which contradicts the proposition of over-pessimism.
> Nonetheless, whether Wasserstein DRO suffers from a similar degradation problem is still an open question, and we want to discover that in future work.
>
> ---
>
> [a]Hu, Weihua, et al. "Does distributionally robust supervised learning give robust classifiers?." International Conference on Machine Learning. PMLR, 2018.

---

> > ### Comment · Area_Chair_D8vA · 2023-08-18
> >
> > Thank you for adding more results though Melanoma is still a small data. One related question here is why Wasserstein DRO? Other techniques such as adversarial training, f-divergence could also improve the generalization. Do authors make comparison to these methods?

---

> > > ### Author Response · Authors · 2023-08-20
> > >
> > > >**Q:** Thank you for adding more results though Melanoma is still a small data. One related question here is why Wasserstein DRO? Other techniques such as adversarial training, f-divergence could also improve the generalization. Do authors make comparison to these methods?
> > >
> > > **A:** Thank you for your feedback!
> > >
> > > Our apologies for not providing a clear introduction in our manuscript. We indeed draw comparisons with methods that employ f-divergence. The competitors we refer to in our paper, specifically **ADVShift** ([a]) and **DROLT** ([b]), utilize KL-divergence, which is a particular instance of f-divergence, as their metric. Based on our experiments, our method outperforms these methods.
> > >
> > > Regarding adversarial training, we introduce PGD as an additional competitor. Following the setting in [c], we generate adversarial examples within an $l_2$-norm ball with a radius of $128/255$ and a step size of $15/255$. We only tune the learning rate on the validation set, and the testing results are shown in the subsequent table.
> > >
> > > **The results of PGD-10:**
> > >
> > > |Datasets|Models|0.01-C|0.05-C|0.10-C|0.20-C|0.01|0.05|0.10|0.20|
> > > |:-:|:-:|:-:|:-:|:-:|:-:|:-:|:-:|:-:|:-:|
> > > |CIFAR10|resnet20|60.86|72.02|79.59|84.82|62.31|75.58|83.54|88.86|
> > > |CIFAR100|resnet20|52.70|58.72|59.50|60.90|53.00|59.84|60.61|62.14|
> > > |CIFAR10|resnet32|60.01|69.13|77.84|83.62|61.14|72.05|81.54|87.31|
> > > |CIFAR100|resnet32|53.05|55.25|58.69|57.30|53.33|55.82|59.67|58.20|
> > > |MNIST|resnet20|96.71|98.17|98.07|98.78|99.59|99.97|99.98|99.99|
> > > |MNIST|small_cnn|94.42|98.21|98.83|98.90|99.53|99.93|99.97|99.98|
> > >
> > > From the results, it is evident that PGD narrows the generalization gap. However, in a long-tailed setting, PGD exhibits suboptimal performance in clean scenarios, which is also potentially attributable to the robust overfitting phenomenon detailed in [c]. Thus, its robust outcomes are not as high as those of our methods.
> > >
> > > [a]Zhang, Jingzhao, et al. "Coping with label shift via distributionally robust optimisation." arXiv preprint arXiv:2010.12230 (2020).
> > >
> > > [b]Samuel, Dvir, and Gal Chechik. "Distributional robustness loss for long-tail learning." Proceedings of the IEEE/CVF International Conference on Computer Vision. 2021.
> > >
> > > [c]Rice, Leslie, Eric Wong, and Zico Kolter. "Overfitting in adversarially robust deep learning." International Conference on Machine Learning. PMLR, 2020.

---

> > ### Comment · Reviewer_9FMm · 2023-08-18
> >
> > Thank the authors for the rebuttal. R1 addresses my concern about the scalability of the proposed method since the results on Tiny-ImageNet-200 are provided. However, I can't entirely agree with R2. Wasserstein DRO also suffers from overly pessimistic issues as indicated by [1]. And f-divergence DRO is an important baseline for evaluation, but the author did not seem to provide the results of f-divergence DRO for comparison.
> >
> > [1] Frogner, Charlie, et al. "Incorporating unlabeled data into distributionally robust learning." arXiv preprint arXiv:1912.07729 (2019).

---

> > > ### Author Response · Authors · 2023-08-20
> > >
> > > >**Q:** Thank the authors for the rebuttal. R1 addresses my concern about the scalability of the proposed method since the results on Tiny-ImageNet-200 are provided. However, I can't entirely agree with R2. Wasserstein DRO also suffers from overly pessimistic issues as indicated by [1]. And f-divergence DRO is an important baseline for evaluation, but the author did not seem to provide the results of f-divergence DRO for comparison.
> > >
> > > **A:** Thank you for your constructive comment! We are glad to hear that your concern regarding scalability has been sufficiently addressed. With respect to f-divergence, we regret the oversight in our manuscript. Notably, our competitors, **DROLT**([a]) and **ADVShift**([b]), employ DRO based on KL-divergence to enhance model robustness, and our methods consistently outperform theirs across all experiments.
> > >
> > > To our knowledge, [c] reveals a phenomenon quite different to that described in [d]. The former suggests that in Wasserstein DRO, the training process may collapse, leading the model to a trivial solution, even when the perturbation radius $\epsilon$ is relatively small compared to the distance between training distribution and true distribution.
> > >
> > > To check whether this anomaly manifests in our setting, we conduct a simple two-step experiment on CIFAR datasets. In the first step, we estimate an upper-bound of Wasserstein distance between training and testing distributions. In the second step, we set $\epsilon$ to be larger than the estimated distance, and see if similar issue occurs in our context.
> > >
> > > **Step 1:** Assume that both training and test sets of the original CIFAR are sampled from the same distribution, and our corrupted CIFAR datasets are sampled from the "real distribution". Directly solving the Wasserstein DRO between these distributions remains intractable. However, given that the corrupted datasets are generated from the original testing sets, a good estimate of the Wasserstein distance can be obtained by calculating the mean example-wise distance between corrupted and clean datasets, resulting in:
> > >
> > > **The estimated Wasserstein distance between CIFAR and different levels of CIFAR-C:**
> > >
> > > |Datasets|Level1|Level2|Level3|Level4|Level5|
> > > |:-:|:-:|:-:|:-:|:-:|:-:|
> > > |CIFAR10|2.80|4.07|4.67|5.58|6.63|
> > > |CIFAR100|2.83|4.11|4.67|5.57|6.61|
> > >
> > > **Step 2:** Train our model with $\epsilon \in \\{1,2,3,4,5,6,7,8\\}$, which is much larger compared to our experimental parameters. In the 16 groups of experiments we run on CIFAR10 and CIFAR100, only $\epsilon=7$ in CIFAR100 fails to converge.  However, by choosing smaller learning rate, our model can evade collapsing into a trivial solution.
> > > This experiment proves that the training collapse problem is not that fatal in our setting, even when $\epsilon$ is substantially larger than the distance to the true data distribution, and can be avoided by simply tuning the learning rate.
> > >
> > > However, shifting the discussion about overly-pessimistic to a broader perspective, beyond a specific setting, we must probe if our methods encounter performance degradation under large, unconstrained uncertainty set. This becomes similar to what we discussed in Prop. 2, i.e., if we place no constraint on the distributional attacker, it will tend to attack the tail class, potentially causing label noise. This might typify the over-pessimistic problem inherent in our setting. Nonetheless, we have proposed our solution by adding structural constraint, namely DRAUC-Da, in our paper.
> > >
> > > [a]Samuel, Dvir, and Gal Chechik. "Distributional robustness loss for long-tail learning." Proceedings of the IEEE/CVF International Conference on Computer Vision. 2021.
> > >
> > > [b]Zhang, Jingzhao, et al. "Coping with label shift via distributionally robust optimisation." arXiv preprint arXiv:2010.12230 (2020).
> > >
> > > [c]Frogner, Charlie, et al. "Incorporating unlabeled data into distributionally robust learning." arXiv preprint arXiv:1912.07729 (2019).
> > >
> > > [d]Hu, Weihua, et al. "Does distributionally robust supervised learning give robust classifiers?." International Conference on Machine Learning. PMLR, 2018.

---

### Official Review · Reviewer_2tJ9 · 2023-07-25

**Soundness:** 3 good
**Presentation:** 2 fair
**Contribution:** 2 fair
**Rating:** 4
**Confidence:** 3

**Summary:**

The authors propose a Distributionally Robust AUC (DRAUC) optimization method that combines Wasserstein distance based DRO with AUC optimization. The combination is designed to be robust for data perturbations. The authors provide theoretical analysis from perturbation and generalization perspective, and justify the practical performance on the perturbed data version for MNIST and CIFAR10.

**Strengths:**

1) The motivation is clear: combining DRO (perturbation robust) with AUC.
2) Authors provide both theoretical and practical justifications.

**Weaknesses:**

1) The idea is straightforward and the contribution is limited. Perturbation robustness idea has been explored for adversarial defense (e.g. Hongyang Zhang, Yaodong Yu, Jiantao Jiao, Eric Xing, Laurent El Ghaoui, and Michael Jordan. Theoretically principled trade-off between robustness and accuracy. In International conference on machine learning, pages 7472–7482. PMLR, 2019; Yin, Dong, Ramchandran Kannan, and Peter Bartlett. "Rademacher complexity for adversarially robust generalization." International conference on machine learning. PMLR, 2019). From my view point, this paper directly extend the perturbation robustness to AUC learning.

2) There are some issues for presentation: i) the $\phi_\lambda(z,\mathbf\theta)$ in line 94 doesn't make sense because the $z'$ will be simply optimized as the value $z$. The problem exists through the whole main content. ii) The '=' should be removed for E.q. 15 to be consistent with other presentations. iii) The $\mathcal W$ below line 144 should be $\mathcal W_c$. iv) The notations for $\hat Q_+$ and $\hat Q_-$ are abused for E.q. 18 and 19, which are confusing. v) Algorithm 1 and Algorithm 2 are similar, which can be further simplified or merged.

3) It might be better to include (D. Zhu, G. Li, B. Wang, X. Wu, and T. Yang. When auc meets dro: Optimizing partial auc for deep learning with non-convex convergence guarantee. In International Conference on Machine Learning, pages 27548–27573. PMLR, 2022) as another baseline for experiments because it also combines DRO with AUC, although it is not designed for perturbation robustness.

4) The Figure 3 is not very informative. I can only observe CE loss is worse than AUC-type losses.

**Questions:**

Please see the weaknesses.

**Limitations:**

Please see the weaknesses.

---

> ### Author Rebuttal · Authors · 2023-08-09
>
> Thank you for your valuable questions! To address your concern, we restate our contributions, fix the typos, and add AUCDRO as a new baseline. The details are as follows:
>
> >**Q1:** The idea is straightforward and the contribution is limited. Perturbation robustness idea has been explored for adversarial defense. From my view point, this paper directly extend the perturbation robustness to AUC learning.
>
> **A1:** Thank you for your inquiry. We want to clarify that the integration of DRO and AUC optimization is not trivial for several reasons:
>
> - **Identifying the strong duality of Distributionally Robust AUC is challenging.** To get rid of the intractable Wasserstein perturbation, one must employ strong duality, as in [a], to reformulate the original optimization problem. Nonetheless, the strong duality becomes much more complicated when assessing performance using AUC, which necessitates a pair-wise loss.
>
> - **The direct integration of DRO with AUC optimization imposes unaffordable computational costs.** Wasserstein DRO utilizes an adversarial training framework to perturb examples. The inner K-step PGD is computationally expensive. The pair-wise formulation requires a time complexity of $O(N^2dK)$ and a spatial complexity of $O(Nd)$, where $N$ is the size of the training set, and $d$ represents the input dimensions. Such complexities make this problem impossible to solve for large datasets. To this end, we conduct an instance-wise reformulation, successfully reducing the time complexity to $O(Ndk)$ and the spatial complexity to $O(Bd)$, where $B$ is the batch size.
> - **The reformulation introduces an intractable optimization problem.** Unfortunately, the original min-max-min-max formulation proves to be intractable. To address this, we propose a surrogate loss which serves as the upper bound of the original objective. This is also specifically designed for our algorithm.
> - **The collision of DRO and long-tailed optimization algorithm results in intriguing outcomes.** Under the assumptions of a neural collapsed feature space, the distributional attacker tends to target the tail classes. In defense of this, we propose distribution-aware DRAUC and evaluate its effectiveness empirically and theoretically.
> - **Typical Rademacher complexity theorem in DRO and adversarial training does not suit our setting.** The vital challenge is that the symmetrization scheme is not available in the pair-wise formulation of AUC. For instance, $\ell(f_{{\theta}}(x_1) -f_{{\theta}}(x_2))$ is interdependent with $\ell(f_{{\theta}}(x_1') -f_{{\theta}}(x_2'))$, if $x_1 = x_1'$ or $x_2 = x_2'$, which makes prior works not suitable in our setting. To address this issue, we propose the Rademacher complexity based on the instance-wise reformulation and successfully solve this problem entirely under the instance-wise setting. Besides, we build control of the excess risk of DRAUC by this Rademacher complexity, which is simpler and easier to use than prior analysis on the Rademacher complexity of AUC.
> ---
>
> >**Q2:** The $\phi_\lambda(z,\theta)$ in line 94 doesn't make sense because the $z'$ will be simply optimized as the value. The problem exists through the whole main content.
>
> **A2:** Thanks for your careful review! The term $\ell(f_\theta,z)$ on the right-hand side should actually be $\ell(f_\theta,z')$, making the corrected equation  $\phi_\lambda(z, \theta) = \sup_{z' \in \mathcal Z}\{\ell(f_\theta,z') - \lambda c(z,z')\}$. We also correct the typos in Line 127 and Line 170.
>
> ---
>
> >**Q3:** The '=' should be removed for E.q. 15 to be consistent with other presentations. The $\mathcal W$ below line 144 should be $\mathcal W_c$.
>
> **A3:** We have corrected the typos. Thanks a lot!
>
> ---
>
> >**Q4:** The notations for and are abused for E.q. 18 and 19, which are confusing.
>
> **A4:** Thank you for pointing out the problem. We will modify our description to define the domain of our perturbation:
>
> $\mathcal Q = \\{Q|~\mathcal W_c(\widehat Q_+,\widehat P_+)\le \epsilon_+, \mathcal W_c(\widehat Q_-,\widehat P_-)\le \epsilon_-\\}$
>
> and denotes as $\max_{Q\in\mathcal Q}$.
>
> ---
>
> >**Q5:** Algorithm 1 and Algorithm 2 are similar, which can be further simplified or merged.
>
> **A5:** Thank you for your valuable suggestion! We have integrated Algorithm 1 and Algorithm 2 by vectorizing the parameters.
>
> ---
>
> >**Q6:** It might be better to include AUCDRO as another baseline for experiments because it also combines DRO with AUC.
>
> **A6:** Thank you for your suggestion! We add this method as our baseline (AUCDRO), and the results are listed as follows:
>
> |Datasets|Models|0.01-C|0.05-C|0.10-C|0.20-C|0.01|0.05|0.10|0.20|
> |:-:|:-:|:-:|:-:|:-:|:-:|:-:|:-:|:-:|:-:|
> |CIFAR10|resnet20|63.35|76.19|81.82|85.96|67.14|84.00|90.92|94.88|
> |CIFAR10|resnet32|65.10|71.23|81.45|86.23|68.69|78.51|90.67|95.07|
> |CIFAR100|resnet20|55.96|61.65|62.67|65.72|57.14|64.74|66.59|70.66|
> |CIFAR100|resnet32|56.93|61.41|64.08|68.93|58.33|64.02|68.71|73.86|
> |MNIST|small_cnn|94.00|97.80|97.76|98.54|99.12|99.82|99.92|99.95|
> |MNIST|resnet20|89.11|94.40|95.71|96.73|98.65|99.87|99.89|99.95|
>
> Though AUCDRO is a state-of-the-art method in partial AUC optimization, it is not designed to improve model robustness; thus, its results on corrupted data are lower than ours.
>
>
> ---
>
> >**Q7:** The Figure 3 is not very informative. I can only observe CE loss is worse than AUC-type losses.
>
> **A7:** Thank you for pointing out this problem! In constructing the long-tailed version datasets, we assign the first half of classes as positive and the rest half as negative. This results in a large intra-class variance, making it challenging for even a well-trained classifier to generate collapsed features. To prevent any misunderstanding, we decide to move this figure to the supplementary materials.
>
> ---
>
> [a]Gao, Rui, and Anton Kleywegt. "Distributionally robust stochastic optimization with Wasserstein distance." Mathematics of Operations Research 48.2 (2023): 603-655.

---

> ### Author Response · Authors · 2023-08-18
>
> Dear reviewer 2tJ9,
>
> We want to kindly inquiry if our response adequately addresses your concerns? If there are any questions, please feel free to ask and we will be glad to clarify.

---

> > ### Comment · Area_Chair_D8vA · 2023-08-18
> >
> > It is not clear why pairwise formulation requires (N^2) complexity.  Since the algorithm can be always stochastic based on mini-batch samples, it do not need all pairs in the data.  For pairs in the mini-batch, it is fairly light compared with model size. The method base on pairwise loss formulation is a natural baseline.

---

> > > ### Author Response · Authors · 2023-08-20
> > >
> > > >**Q:** It is not clear why pairwise formulation requires (N^2) complexity. Since the algorithm can be always stochastic based on mini-batch samples, it do not need all pairs in the data. For pairs in the mini-batch, it is fairly light compared with model size. The method base on pairwise loss formulation is a natural baseline.
> > >
> > > **A:** We appreciate your insightful feedback. The primary challenge associated with the pairwise formulation lies in generating the local worst distribution. The pairwise formulation of DRAUC is defined as:
> > >
> > > ${DRAUC}\_\epsilon(f\_\theta, \widehat P) = 1 -\max\_{\widehat Q:\mathcal W\_c(\widehat Q,\widehat P) \le \epsilon} \mathbb E\_{\widehat Q} [{\ell(f\_\theta(x^+)-f\_\theta(x^-))}]$
> > >
> > > Generating the local-worst distribution $\widehat Q $ suffers from two main difficulties. Firstly, as highlighted in our response to reviewer 2tJ9, the strong duality becomes very complicated when dealing with pairwise loss. Consequently, the Wasserstein perturbation becomes nearly infeasible. Secondly, generating $\widehat Q $ is loss-dependent, implying that we need to know all the training details to deliver a malicious attack. In our endeavor to secure a performance guarantee for our model, we cannot limit the scope of information accessible to an attacker. This results in a computational cost remaining at  $O(N^+N^-dK)$, even within a stochastic setting. We will provide further explanations regarding the unsuitability of the pairwise formulation in Sec 3.1.
> > >
> > > While the pairwise formulation might not be ideal for a distributionally robust setting, it can be considered as a baseline of natural training. The outcomes are presented as below:
> > >
> > > **The results of pairwise AUC(PAUC):**
> > >
> > > |Datasets|Models|0.01-C|0.05-C|0.10-C|0.20-C|0.01|0.05|0.10|0.20|
> > > |:-:|:-:|:-:|:-:|:-:|:-:|:-:|:-:|:-:|:-:|
> > > |CIFAR10|resnet20|56.31|75.75|82.69|84.07|57.43|84.35|90.94|94.30|
> > > |CIFAR10|resnet32|61.81|75.24|80.44|84.06|65.01|81.46|89.90|94.21|
> > > |CIFAR100|resnet20|52.15|61.19|64.98|69.64|52.58|63.24|67.52|73.40|
> > > |CIFAR100|resnet32|53.52|61.72|65.21|69.17|54.24|63.65|67.73|73.30|
> > > |MNIST|small_cnn|94.58|97.24|98.39|98.90|99.14|99.86|99.92|99.96|
> > > |MNIST|resnet20|90.28|92.68|90.35|98.65|98.65|99.87|99.89|99.95|

---

> > ### Comment · Reviewer_2tJ9 · 2023-08-18
> > **Thanks for your rebuttal**
> >
> > The reviewer has read the rebuttal and appreciate the efforts made by the authors. However, the reviewer still holds the original evaluation score. Although my previous concerns about the writing issues can be fixed, I don't feel the paper is well prepared by the deadline of NeurIPS. Moreover, the experimental results don't include variance or standard deviation and might not be repeated independently multiple times.

---

> > > ### Author Response · Authors · 2023-08-20
> > >
> > > >**Q:** The reviewer has read the rebuttal and appreciate the efforts made by the authors. However, the reviewer still holds the original evaluation score. Although my previous concerns about the writing issues can be fixed, I don't feel the paper is well prepared by the deadline of NeurIPS. Moreover, the experimental results don't include variance or standard deviation and might not be repeated independently multiple times.
> > >
> > > **A:** Thank you for your comment! We are glad to hear that our rebuttal has well addressed your previous concerns. Thanks to the valuable suggestions the reviewers proposed, our manuscript has become more complete and convincing; we really appreciate that!
> > > Regarding the matter of the error bar, we would like to clarify that due to the considerable computational expense, it is widely accepted within the realms of Wasserstein DRO and adversarial training to refrain from repeated experiments, as evidenced by references [a, b, c, d]. Nevertheless, in order to allay your concerns, we have repeated our method on CIFAR10 three times using varied seeds and present the average results and their standard deviation:
> > >
> > > **The results were repeated 3 times with different seeds on CIFAR10 using ResNet20:**
> > > |Methods|0.01-C|0.05-C|0.10-C|0.20-C|0.01|0.05|0.10|0.20|
> > > |:-:|:-:|:-:|:-:|:-:|:-:|:-:|:-:|:-:|
> > > |DRAUC|65.18(1.07)|79.87(0.64)|85.78(0.48)|89.35(0.13)|68.05(0.69)|86.19(0.16)|90.78(0.37)|94.38(0.20)|
> > > |CDRAUC|65.43(0.18)|79.89(0.55)|85.74(0.41)|89.62(0.31)|67.79(0.50)|84.37(0.65)|90.69(0.34)|94.3(0.19)|
> > >
> > > The results indicate that the variances of our method are small.
> > >
> > > [a]Sinha, Aman, et al. "Certifying some distributional robustness with principled adversarial training." arXiv preprint arXiv:1710.10571 (2017).
> > >
> > > [b]Bui, Tuan Anh, et al. "A unified wasserstein distributional robustness framework for adversarial training." arXiv preprint arXiv:2202.13437 (2022).
> > >
> > > [c]Madry, Aleksander, et al. "Towards deep learning models resistant to adversarial attacks." arXiv preprint arXiv:1706.06083 (2017).
> > >
> > > [d]Jiang, Ziyu, et al. "Robust pre-training by adversarial contrastive learning." Advances in neural information processing systems 33 (2020): 16199-16210.

---

### Author Rebuttal · Authors · 2023-08-09

Dear reviewers,
we sincerely appreciate your time and valuable comments. Thanks to your detailed, insightful reviews, we introduce the following changes:

- **Additional Empirical Support:** We add AUCDRO[a] as a new baseline. We conduct experiments on **large-scale datasets** including Melanoma and Tiny-ImageNet-200. We also evaluate our proposed method on **modern network architectures** including EfficientNet and DenseNet.
- **Method Expansion:** We extend our algorithm to multi-class classification scenarios.
- **Improvement on readability:** We combine the DRAUC-Df/Da algorithm together for a more streamlined presentation, and fix several typos in our paper due to your careful reviews. We appreciate the thorough review.

**Empirical results:**
- **New baseline:**
  - **AUCDRO([a]):** As mentioned by Reviewer 2tJ9, it is better to compare with AUCDRO due to its integration of DRO and AUC optimization (though it is not designed for perturbed defense).
- **New datasets:**
  - **Tiny-ImageNet-200**: This dataset comprises 110,000 images in 200 classes, with a resolution of 64x64x3. We assign the binary long-tailed version by utilizing the hyper-class information. As detailed, we construct three subsets, Tiny-ImageNet-Dogs, Birds and Vehicles. Model robustness is then assessed on Tiny-ImageNet-C([b]).
  - **Melanoma:** Melanoma is a large-scale medical image dataset that contains 32,542 positive examples and 584 negative examples. We subdivided the dataset into training, validation, and testing sets in a 7:1.5:1.5 ratio. Due to our use of K-PGD to generate perturbed examples, running experiments on a large-scale dataset is really computationally expensive. We downsampling all images to a resolution of 224x224, and train our model with a batch size 32.
    To our knowledge, no corrupted version of the Melanoma dataset exists. In assessing the robustness of our proposed algorithm, we created a corrupted version like that presented in [b]. Additionally, we excluded weather-based corruptions such as snow, frost, and fog, which are implausible in the context of medical images. As a result, we incorporated 12 corruptions across five perturbation levels in the testing split of the Melanoma dataset to generate Melanoma-C.
  - **Multi-class CIFAR10-LT:** Multi-class CIFAR10-LT is a long-tailed version sampled from the original CIFAR10. The degree of class imbalance is indicated by the imratio, defined as the ratio of the smallest class's size to the largest. We conduct similar reformulation to AUCMLoss to compute multi-class AUC.

- **New backbones:**
  - **Efficientnet, Densenet121**.

**Methodlogically results:** We expand our DRAUC-Df and DRAUC-Da to multi-class settings, the derivation is as follows:

First, we define multi-class AUC:

$AUCM(f\_{{\theta}}) = \frac{1}{N\_C}\sum\_{i=1}^{N\_C}AUC\_i(f\_{{\theta}}) = \frac{1}{N\_C}\sum\_{i=1}^{N\_C} \mathbb E\_{\widehat P\_i, {\widehat P\_i'}}[\ell(f\_{{\theta}}(x\_i) -f\_{{\theta}}(x\_i'))]$

where $N\_C$ is the number of classes, $AUC\_i$ denotes the AUC score when assigning $i^{th}$ class as positive, and the rest classes as negative, $\widehat P\_i$ denotes the example distribution of $i^{th}$ class and $\widehat P\_i'$ denotes the distribution or the rest classes. Consistent with the notation in our paper, we use $\widehat Q$ to describe the perturbed distribution. Following the idea of optimizing the multi-class AUC under local worst distribution, we now present the objective of Multi-class DRAUC(MDRAUC)

$\min\_{{\theta}}\max\_{\widehat Q:\mathcal W\_c(\widehat Q,\widehat P)\le \epsilon}  ~\mathbb E\_{\widehat Q}\left[\frac 1{N\_C}\sum\_{i=1}^{N\_C}AUC\_i(f\_{{\theta}})\right]$

Upon applying the instance-wise reformulation, we obtain

$\min\_{{\theta}}\max\_{\widehat Q:\mathcal W\_c(\widehat Q,\widehat P)\le \epsilon} \frac 1{N\_C}\sum\_{i=1}^{N\_C}\min\_{{(a\_i,b\_i)\in \Omega\_{a,b}}}\max\_{{\alpha\_i \in \Omega\_{\alpha} }} ~\mathbb E\_{\widehat Q}\left[g\_i(a\_i,b\_i,\alpha\_i,{{\theta}},z)\right]$

where

$g\_i(a\_i,b\_i,\alpha\_i,\theta,{z})= (1-p\_i)\cdot (f\_\theta({x}) - a\_i) ^ 2 \cdot \mathbb I\_{[y = i]} + p\_i \cdot (f\_\theta( x) - b\_i) ^ 2 \cdot  \mathbb I\_{[y \ne i]} $

$\qquad\qquad\qquad\qquad\qquad+ 2\cdot(1+\alpha\_i)\cdot(p\_i \cdot f\_\theta( x) \cdot \mathbb I\_{[y \ne i]} - (1-p\_i)\cdot f\_\theta( x) \cdot \mathbb I\_{[y = i]} - p\_i(1-p\_i)\cdot\alpha\_i^2).$

Here, $p\_i = \frac{n\_i} N$ denotes the proportion of examples in the $i^{th}$ class. Moreover, with ${\theta}$ and $\widehat Q$ fixed, $a\_i,a\_j$ are irrelevant $\forall i,j$, as are $b$ and $\alpha$. Thus, we can safely interchange the summation and inner min-max, resulting in

$\min\_{{\theta}}\max\_{\widehat Q:\mathcal W\_c(\widehat Q,\widehat P)\le \epsilon} \min\_{{(\vec a,\vec b)}}\max\_{{\vec\alpha }} ~\mathbb E\_{\widehat Q}\left[\frac 1{N\_C}\sum\_{i=1}^{N\_C}g\_i(a\_i,b\_i,\alpha\_i,{{\theta}},z)\right]$

where $\vec a = \\{a\_1,...a\_{N\_C}\\},\vec b = \\{b\_1,...b\_{N\_C}\\},\vec \alpha = \\{\alpha\_1,...\alpha\_{N\_C}\\}$. Using a technical similar to Sec 3.2, we obtain  our formulation of MDRAUC-Df as follows

${(MDf\star)}\quad\min\_{{{\theta}},\vec a,\vec b} \min\_{\lambda > 0}\max\_{\vec \alpha} \left\\{\lambda\epsilon + \mathbb E\_{\widehat{ P}}[\widehat \varphi\_{{{\theta}}, \vec a, \vec b, \vec \alpha, \lambda}(z)] \right\\}$

and

$\widehat \varphi\_{{{\theta}}, \vec a, \vec b, \vec \alpha, \lambda}(z) = \max\_{z' \in \mathcal Z}\left[\frac 1{N\_C}\sum\_{i=1}^{N\_C}g\_i(a\_i,b\_i,\alpha\_i,{{\theta}},z) - \lambda c(z,z')\right].$

---

[a]Zhu, Dixian, et al. "When auc meets dro: Optimizing partial auc for deep learning with non-convex convergence guarantee." International Conference on Machine Learning. PMLR, 2022.

[b]Hendrycks, Dan, and Thomas Dietterich. "Benchmarking neural network robustness to common corruptions and perturbations." arXiv preprint arXiv:1903.12261 (2019).

---

### Decision · Program_Chairs · 2023-09-21

**Decision:**

Accept (poster)

**Comment:**

This paper studies AUC maximization for distributional shift. The paper incorporates Wasserstein distance based DRO into min-max formulation of AUC maximization and derives upper bound of the objective for better optimization.

No critical issues have been raised in the reviews and during the discussion phase. The authors are encouraged to make the following changes in their revision.

The claim about training costs of pairwise loss formulation is not correct. Similar DRO technique could be easily injected into pairwise loss formulation and the optimization could be much easier. More discussion and comparison should be done.

Related work should be moved to the main text given there are tremendous literature on AUC maximization and DRO.

The distributional shift should be made clear in the experiments. Other strong baselines about AUC maximization that should be compared, e.g., SOPA-s, Compositional AUC [1].

[1] Compositional Training for End-to-End Deep AUC Maximization. Yuan et al.